YITP-SB-2024-09

# Non-invertible and higher-form symmetries in 2+1d lattice gauge theories

Yichul Choi[1,2], Yaman Sanghavi[1], Shu-Heng Shao[1], and Yunqin Zheng[1]

[1]*C. N. Yang Institute for Theoretical Physics, Stony Brook University*
[2]*Simons Center for Geometry and Physics, Stony Brook University*

## Abstract

We explore exact generalized symmetries in the standard 2+1d lattice $\mathbb{Z}_2$ gauge theory coupled to the Ising model, and compare them with their continuum field theory counterparts. One model has a (non-anomalous) non-invertible symmetry, and we identify two distinct non-invertible symmetry protected topological phases. The non-invertible algebra involves a lattice condensation operator, which creates a toric code ground state from a product state. Another model has a mixed anomaly between a 1-form symmetry and an ordinary symmetry. This anomaly enforces a nontrivial transition in the phase diagram, consistent with the "Higgs=SPT" proposal. Finally, we discuss how the symmetries and anomalies in these two models are related by gauging, which is a 2+1d version of the Kennedy-Tasaki transformation.

# 1  Introduction

An important class of problems in physics is to understand the low energy phase diagram of a given quantum system, which can be a continuum quantum field theory or a lattice model. One powerful non-perturbative tool is the use of global symmetries and anomalies. In the past decade, the notion of global symmetries has been generalized in many different directions,

including the higher-form and non-invertible symmetries. These generalized symmetries provide novel constraints on the phase diagrams and lead to new selection rules in field theories and lattice models. See [1–8] for reviews.

There have been extensive works on lattice realizations of non-invertible symmetries in 1+1d.[1] The most well-studied example in 1+1d is the Kramers-Wannier duality symmetry in the Ising model [9–16]. In particular, it has been realized recently that on a tensor product Hilbert space, the operator algebra of the non-invertible Kramers-Wannier operator mixes with lattice translation [15, 16].[2] Furthermore, it implies a Lieb-Schultz-Mattis-type constraint [16] (see also [25]), which is reminiscent of the constraint from 't Hooft anomalies. On the other hand, gapped topological phases protected by (non-anomalous) non-invertible symmetries have been realized in [26–29]. See, for example, [30–45] for recent discussions of other lattice non-invertible symmetries in 1+1d.

In this work, we explore exact non-invertible and high-form global symmetries in conventional 2+1d lattice models, and discuss their consequences on the phase diagram. We find that these novel symmetries exist ubiquitously in familiar models.

## 1.1   Lattice $\mathbb{Z}_2$ gauge theory coupled to Ising matter

We consider a simple class of lattice models —the lattice $\mathbb{Z}_2$ gauge theory coupled to Ising matter [46–53].

We focus on 2+1d translational invariant Hamiltonian systems on a periodic square lattice with $L_x$ and $L_y$ sites in the $x$ and $y$ directions, respectively. On each site $v$, there is a qubit representing the Ising matter degrees of freedom. The local site Hilbert space is $\mathcal{H}_v \cong \mathbb{C}^2$, acted upon by Pauli $X_v$ and $Z_v$. On each link $\ell$, there is also a qubit representing the $\mathbb{Z}_2$ gauge degrees of freedom. The local link Hilbert space is $\mathcal{H}_\ell \cong \mathbb{C}^2$, acted upon by Pauli $\sigma_\ell^x$ and $\sigma_\ell^z$. The Hilbert space is a tensor product of the local site and link Hilbert spaces $\mathcal{H} = \bigotimes_v \mathcal{H}_v \otimes \bigotimes_\ell \mathcal{H}_\ell$.[3] A generic Hamiltonian is a sum of local operators composed of $X_v, Z_v, \sigma_\ell^x, \sigma_\ell^z$ in a finite region. Below, we enumerate a few familiar examples.

---

[1]Throughout this work, we focus on quantum lattice models, as opposed to statistical/Euclidean lattice models. (We often omit the word "quantum" for brevity.) By a $(D+1)$-dimensional quantum lattice model we mean that the spatial lattice is $D$-dimensional.

[2]In 1+1d, a general fusion category symmetry can be realized on an anyonic chain without mixing with the lattice translation [17, 18]. However, the anyonic chain generally does not have a tensor product Hilbert space. See [19] for a recent generalization of the anyonic chain to 2+1d. See also [20, 21] for the realization of non-invertible symmetries in 2+1d spin models. Another generalization of non-invertible symmetries to 2+1d is related to the subsystem symmetries [22–24].

[3]In other words, we work on a Lieb lattice.

**Ising model** The Hamiltonian for the Ising model only involves the qubits on the sites:

$$H_{\text{Ising}} = -h \sum_v X_v - J \sum_{\langle v,v' \rangle} Z_v Z_{v'} \,, \tag{1.1}$$

where the sum in $\langle v, v' \rangle$ is over all neighboring vertices $v, v'$. The theory has an ordinary $\mathbb{Z}_2^{(0)}$ (0-form) symmetry,[4] generated by $U = \prod_v X_v$. Its phase diagram depends on $J/h$.[5] When $J/h \to 0$, the theory is in the trivially gapped phase where $\mathbb{Z}_2^{(0)}$ is unbroken. When $J/h \to \infty$, the theory is in the $\mathbb{Z}_2^{(0)}$ spontaneously broken phase. There is a second order phase transition around $J/h \sim 0.33$ [55].

**Lattice $\mathbb{Z}_2$ gauge theory** The Hamiltonian for the lattice $\mathbb{Z}_2$ gauge theory only involves the qubits on the links [56]:

$$H_{\text{gauge}} = -\tilde{h} \sum_v \prod_{\ell \ni v} \sigma_\ell^x - g \sum_f \prod_{\ell \in f} \sigma_\ell^z - \tilde{J} \sum_\ell \sigma_\ell^z \,. \tag{1.2}$$

The first term is the Gauss law term, which energetically suppresses electric charge on each vertex $v$.[6] The second term is the magnetic flux term, which energetically suppresses magnetic flux around each plaquette $f$. The first two terms commute with each other, and are the local terms of the toric code Hamiltonian $H_{\text{TC}}$ [57], which in the low energy describes the topological, continuum $\mathbb{Z}_2$ gauge theory. The last term $\sigma_\ell^z$ creates virtual pairs of $\mathbb{Z}_2$ charges, which condense in the large $\tilde{J}$ limit.

The lattice $\mathbb{Z}_2$ gauge theory can be obtained by gauging the $\mathbb{Z}_2^{(0)}$ symmetry of the Ising model, hence we also refer to it as the gauged Ising model and denote it as $\frac{\text{Ising}}{\mathbb{Z}_2}$.

The Hamiltonian has a conserved operator $\eta(\gamma) = \prod_{\ell \in \gamma} \sigma_\ell^z$ supported along any closed loop $\gamma$. It generates a $\mathbb{Z}_2^{(1)}$ 1-form global symmetry. We will impose this symmetry throughout, which forbids the symmetry-breaking term $\sum_\ell \sigma_\ell^x$. This more general model with $\sum_\ell \sigma_\ell^x$ has been studied extensively in [46–48, 50–52, 49, 53].

One can alternatively view the magnetic flux term as the Gauss law for the magnetic gauge field on the dual lattice. The two viewpoints are related by the electro-magnetic duality symmetry in the toric code and $\mathbb{Z}_2$ gauge theory.

---

[4]The superscript $(q)$ denotes the form degree of a $q$-form global symmetry [54]. For 0-form symmetries (i.e., ordinary symmetries), we sometimes omit the superscript (0).

[5]Throughout this paper, all parameters $h, J$, etc. are assumed to be non-negative.

[6]Strictly speaking, a lattice gauge theory has the Gauss law imposed strictly as an operator equation, and the Hilbert space is not a tensor product. In this paper, we consider more general models where the Gauss law is imposed energetically, and the Hilbert space is a tensor product. We still loosely refer to them as "lattice gauge theories."

**Lattice $\mathbb{Z}_2$ gauge theory coupled to Ising matter**   Next, we add the Hamiltonians for the Ising model and the lattice $\mathbb{Z}_2$ gauge theory together, and couple them by the $\mathbb{Z}_2^{(0)} \times \mathbb{Z}_2^{(1)}$ symmetric terms in the second line below:

$$
\begin{aligned}
H_{\text{Ising+gauge}} = &- h \sum_v X_v - J \sum_{\langle v,v' \rangle} Z_v Z_{v'} - \tilde{h} \sum_v \prod_{\ell \ni v} \sigma_\ell^x - g \sum_f \prod_{\ell \in f} \sigma_\ell^z - \tilde{J} \sum_\ell \sigma_\ell^z \\
&- h' \sum_v X_v \prod_{\ell \ni v} \sigma_\ell^x - J' \sum_{\langle v,v' \rangle} Z_v \sigma_{\langle v,v' \rangle}^z Z_{v'} + \cdots .
\end{aligned}
\tag{1.3}
$$

The $h'$ term enforces the (electric) Gauss law coupled to Ising matter energetically.[7] This model has a rich phase diagram as one varies the parameters. Below, we will focus on two special corners of the phase diagram, and explore the generalized symmetries, including the higher-form symmetry and non-invertible symmetries.

Apart from the Ising model and lattice $\mathbb{Z}_2$ gauge theory mentioned above, two more interesting special cases of (1.3) are the Fradkin-Shenker model at $J = \tilde{h} = \tilde{J} = 0$ [47, 49],[8] and the cluster model at $h = J = \tilde{h} = g = \tilde{J} = 0$ [58, 49].

## 1.2   Lattice non-invertible symmetry

The first example, analyzed in Section 2, is obtained by restricting the parameters in the Hamiltonian (1.3) to:

$$
h = \tilde{h} , \qquad J = \tilde{J} .
\tag{1.4}
$$

Then there is an additional symmetry swapping the pairs of terms with the same coefficients, i.e.,

$$
X_v \longleftrightarrow \prod_{\ell \ni v} \sigma_\ell^x , \quad Z_v Z_{v'} \longleftrightarrow \sigma_{\langle v,v' \rangle}^z .
\tag{1.5}
$$

We show that this transformation cannot be implemented by an invertible operator. Rather, it is implemented by a *non-invertible* duality operator that commutes with the Hamiltonian:

$$
\mathsf{D} = \frac{1}{2} \left( \prod_{v \neq v_0} \mathsf{S}_v \right) \mathsf{S}_{v_0} \mathsf{C} ,
\tag{1.6}
$$

---

[7]When such Gauss law is enforced strictly by taking $h' \to \infty$, some terms in (1.3) are forbidden, e.g., $J = \tilde{J} = 0$.

[8]More precisely, the Fradkin-Shenker model has the electric Gauss law enforced strictly, i.e., $h' \to \infty$, and contains an additional term $\sum_\ell \sigma_\ell^x$ which breaks 1-form symmetry.

where we have defined the condensation operator as

$$\mathsf{C} = (1 + U) \times \frac{1}{2^A} \sum_\gamma \eta(\gamma) \,. \tag{1.7}$$

Here $\mathsf{S}_v = \frac{1}{2}(1 + G_v + Z_v W_{v_0,v}(1 - G_v))$ is a unitary operator for $v \neq v_0$, where $v_0$ is an arbitrary reference site, $W_{v_0,v}$ is a product of $\sigma_\ell^z$ along a prescribed curve connecting $v_0$ and $v$, and $G_v = X_v \prod_{\ell \ni v} \sigma_\ell^x$. Despite the appearance of a reference point $v_0$, we show that the operator $\mathsf{D}$ is independent of the choice of $v_0$, and is translationally invariant.

These operators obey the algebra

$$\begin{aligned}
\mathsf{D}^2 &= \mathsf{C}, \quad \mathsf{C}^2 = 4\mathsf{C}, \quad \mathsf{D}\mathsf{C} = \mathsf{C}\mathsf{D} = 4\mathsf{D}, \\
\eta\mathsf{D} &= \mathsf{D}\eta = U\mathsf{D} = \mathsf{D}U = \mathsf{D}, \quad \eta\mathsf{C} = \mathsf{C}\eta = U\mathsf{C} = \mathsf{C}U = \mathsf{C}.
\end{aligned} \tag{1.8}$$

This lattice algebra matches the algebra of a fusion 2-category [59]. More specifically, it is 2-Rep$((\mathbb{Z}_2^{(1)} \times \mathbb{Z}_2^{(1)}) \rtimes \mathbb{Z}_2^{(0)})$ which was discussed in [60, 61] in the context of continuum quantum field theory. This is one of the simplest non-invertible symmetries in 2+1d realized in a familiar Hamiltonian model with a tensor product Hilbert space, and our construction is different from the ones in [15, 16, 28].[9]

## 1.3 Lattice 1-form symmetry and its anomaly

Our second example, analyzed in Section 3, corresponds to setting

$$h = h', \qquad J' = \tilde{J}. \tag{1.9}$$

in the Hamiltonian (1.3). Then there is an additional invertible $\mathbb{Z}_{2,\mathrm{swap}}^{(0)}$ 0-form symmetry that swaps the operators

$$X_v \leftrightarrow X_v \prod_{\ell \ni v} \sigma_\ell^x, \qquad Z_v \sigma_{\langle v,v' \rangle}^z Z_{v'} \leftrightarrow \sigma_{\langle v,v' \rangle}^z, \tag{1.10}$$

which can be implemented by an unitary operator

$$V = \prod_v \frac{1}{2} \left( (1 + Z_v) + (1 - Z_v) \prod_{\ell \ni v} \sigma_\ell^x \right). \tag{1.11}$$

---

[9]In Appendix D, we apply this new construction to 1+1d and provide an alternative expression for the non-invertible operator in [28].

This operator creates a $\mathbb{Z}_2^{(0)} \times \mathbb{Z}_2^{(1)}$ symmetry protected topological (SPT) state of [58, 49] from a product state. The two 0-form symmetries $U, V$ and the 1-form symmetry $\eta$ have a mixed anomaly, which we discuss in detail in Section 3.

**Higgs=SPT**  The phase diagram of the Hamiltonian (1.3) has two regions that are both gapped with one ground state and no long-range entanglement. Nonetheless, the two regions are separated by the locus with an anomaly, implying that there must be at least one phase transition between these phases. This is consistent with the observation in [49] that the two phases represent distinct SPT phases.

## 1.4   Non-invertible SPT phases

The non-invertible symmetry $\mathsf{D}$ (along with $\mathbb{Z}_2^{(0)} \times \mathbb{Z}_2^{(1)}$) is non-anomalous in the sense that it can be realized in a gapped phase with a unique ground state. One example is the 2+1d cluster model (4.1) in [58, 49]. Hence, the cluster Hamiltonian is not only a topological phase protected by $\mathbb{Z}_2^{(0)} \times \mathbb{Z}_2^{(1)}$, but also by a non-invertible symmetry $\mathsf{D}$.

We further find another distinct non-invertible SPT phase, described by the following exactly solvable model,

$$H_{\text{cluster}'} = \sum_v X_v \prod_{\ell \ni v} \sigma_\ell^x - \sum_{\langle v, v' \rangle} Y_v \sigma_{\langle v, v' \rangle}^z Y_{v'} - \sum_{\langle v, v' \rangle} Z_v \left( \prod_{\ell \ni v} \sigma_\ell^x \right) \sigma_{\langle v, v' \rangle}^z \left( \prod_{\ell' \ni v'} \sigma_{\ell'}^x \right) Z_{v'} \,. \quad (1.12)$$

We refer to it as the cluster$'$ Hamiltonian. This is a 2+1d generalization of the 1+1d non-invertible SPT phases in [28].

## 1.5   Outline

This paper is organized as follows. In Section 2, we discuss a special class of models in (1.3) with an exact non-invertible symmetry. Section 2.3 discusses the lattice condensation operator and its relation to the toric code ground states. Section 3 discusses a different class of models which realizes an exact 1-form symmetry with a mixed anomaly with other ordinary symmetries. In Section 3.4 we relate this anomaly to the "Higgs=SPT" proposal. In Section 4, we consider 2+1d SPT phases protected by the (non-anomalous) non-invertible symmetry of Section 2. We identify two distinct non-invertible SPT phases, one of them being the cluster model with a 1-form symmetry.

Appendices A and B discuss the continuum field theory counterparts of the non-invertible and 1-form symmetries in Sections 2 and 3, respectively. In Appendix C, we review the

gauging of 0- and 1-form symmetries in 2+1d lattice systems, and use the Ising model as a prototypical example. In Appendix D, we apply our construction in Section 2 to reproduce the $\mathrm{Rep}(D_8)$ non-invertible symmetry in 1+1d of [28]. Appendix E provides more details on the non-invertible operator of Section 2, and Appendix F discusses the relation between the cluster SPT phase and the mixed anomaly involving the 1-form symmetry.

# 2 Lattice non-invertible symmetry

## 2.1 Hamiltonian and symmetries

We begin with the general Hamiltonian (1.3) for the lattice $\mathbb{Z}_2$ gauge theory coupled to Ising matter. We first analyze its symmetries and the corresponding conserved operators that commute with the Hamiltonian.

$\mathbb{Z}_2^{(0)}$ **Ising spin-flip symmetry**   There is an obvious $\mathbb{Z}_2^{(0)}$ 0-form symmetry, whose symmetry operator is

$$U = \prod_v X_v \,, \qquad U^2 = 1 \,, \tag{2.1}$$

which flips $Z_v$ to $-Z_v$ at every site. In Section 3, we will discuss its defects.

**Non-topological vs. topological $\mathbb{Z}_2^{(1)}$ 1-form symmetries**   There is a $\mathbb{Z}_2^{(1)}$ 1-form symmetry operator $\eta(\gamma)$:

$$\eta(\gamma) = \prod_{\ell \in \gamma} \sigma_\ell^z \,, \qquad \eta(\gamma)^2 = 1 \,, \tag{2.2}$$

where $\gamma$ is an arbitrary closed loop. This is an on-site symmetry with no 't Hooft anomaly by itself [62]. This 1-form symmetry should be viewed as a magnetic symmetry, whereas the electric 1-form symmetry $\prod_{\ell \in \widehat{\gamma}} \sigma_\ell^x$ along a $\widehat{\gamma}$ loop on the dual lattice is broken by the coupling to the (electric) matter field and the term $\sum_\ell \sigma_\ell^z$.[10]

In general, this magnetic 1-form symmetry is not topological, in the sense that $\eta(\gamma)$ depends on the detailed shape of the loop $\gamma$. We refer to it as a *non-topological 1-form symmetry*. Clearly, the 0- and 1-form symmetry operators commute.

---

[10]Throughout this paper, we use hatted symbols for lines and surfaces in the dual lattice, e.g. $\widehat{\gamma}, \widehat{\Sigma}$, and unhatted symbols for those in the original lattice, e.g. $\gamma, \Sigma$.

If we take $g \to \infty$ to impose

$$B_f = \prod_{\ell \in f} \sigma_\ell^z = \sigma^z \boxed{\begin{matrix} \sigma^z \\ \\ \sigma^z \end{matrix}} \sigma^z = 1 \tag{2.3}$$

strictly as an operator equation, then the Hilbert space is subject to these constraints and is no longer a tensor product. We denote this constrained Hilbert space as $\widetilde{\mathcal{H}}$. The constraint (2.3) can be interpreted as the Gauss law on the dual lattice for the magnetic gauge field. Once the magnetic Gauss law is imposed strictly, $\eta(\gamma)$ becomes a *topological 1-form symmetry* in the sense that $\eta(\gamma)$ depends only on the homology class of $\gamma$. That is,

$$\eta(\gamma) = \eta(\gamma') \ \ \text{on} \ \ \widetilde{\mathcal{H}}, \quad \text{if} \ \ \gamma \sim \gamma', \tag{2.4}$$

where $\sim$ means that $\gamma, \gamma'$ are in the same homology class. The difference between topological and non-topological 1-form symmetries have been discussed in [63, 64].[11]

There is an important physical distinction between the topological and non-topological 1-form symmetries. A topological 1-form symmetry cannot be broken by any local operators acting within the constrained Hilbert space $\widetilde{\mathcal{H}}$ where the magnetic Gauss law is enforced strictly. This is similar to the continuum 1-form symmetry. However, a non-topological 1-form symmetry can be explicitly broken by a local perturbation, such as $\sum_\ell \sigma_\ell^x$.

**Hint of a non-invertible symmetry** When $h = \tilde{h}, J = \tilde{J}$ in (1.3), there is an additional interesting symmetry. The Hamiltonian is

$$\begin{aligned} H_{\text{noninv}} = &- h \sum_v X_v - J \sum_{\langle v,v' \rangle} Z_v Z_{v'} - h \sum_v \prod_{\ell \ni v} \sigma_\ell^x - g \sum_f \prod_{\ell \in f} \sigma_\ell^z - J \sum_\ell \sigma_\ell^z \\ &- h' \sum_v X_v \prod_{\ell \ni v} \sigma_\ell^x - J' \sum_{\langle v,v' \rangle} Z_v \sigma_{\langle v,v' \rangle}^z Z_{v'} \, . \end{aligned} \tag{2.5}$$

To begin with, we assume the magnetic Gauss law is enforced strictly by sending $g \to \infty$ and work in the constrained Hilbert space $\widetilde{\mathcal{H}}$. Then the Hamiltonian (2.5) is invariant under the following transformation:

$$X_v \leftrightsquigarrow \prod_{\ell \ni v} \sigma_\ell^x, \quad Z_v Z_{v'} \leftrightsquigarrow \sigma_{\langle v,v' \rangle}^z \, . \tag{2.6}$$

---

[11]In [63], the topological/non-topological 1-form symmetries were called non-faithful/faithful, respectively. In [64], they were referred to as the relativistic/non-relativistic 1-form symmetries. We adopt the terminology topological/non-topological symmetries because it directly reflects the property of the symmetry operators.

This transformation is an automorphism of the algebra of operators invariant under $\mathbb{Z}_2^{(0)} \times \mathbb{Z}_2^{(1)}$ generated by $U, \eta$.

However, this transformation cannot be implemented by an invertible operator on a closed periodic lattice for the following reason. Suppose there were an invertible operator $R$ such that $RX_v R^{-1} = \prod_{\ell \ni v} \sigma_\ell^x$. But then $RUR^{-1} = R\left(\prod_v X_v\right) R^{-1} = \prod_v \prod_{\ell \ni v} \sigma_\ell^x = 1$, which is a contradiction.

As we will explain in the rest of the section, the precise meaning of (2.6) is that there is a *non-invertible* operator $\mathsf{D}$ that obeys

$$\mathsf{D}X_v = \left(\prod_{\ell \ni v} \sigma_\ell^x\right)\mathsf{D}, \qquad \mathsf{D}\prod_{\ell \ni v}\sigma_\ell^x = X_v\mathsf{D},$$

$$\mathsf{D}Z_vZ_{v'} = \sigma_{\langle v,v'\rangle}^z\mathsf{D}, \qquad \mathsf{D}\sigma_{\langle v,v'\rangle}^z = Z_vZ_{v'}\mathsf{D}. \tag{2.7}$$

Furthermore, thanks to

$$\mathsf{D}\prod_{\ell \in f}\sigma_\ell^z = \left(\prod_{\ell \in f}\sigma_\ell^z\right)\mathsf{D} = \mathsf{D}, \tag{2.8}$$

we can extend $\mathsf{D}$ to be an operator on the tensor product Hilbert space $\mathcal{H}$, and it commutes with the Hamiltonian, i.e., $\mathsf{D}H_{\mathrm{noninv}} = H_{\mathrm{noninv}}\mathsf{D}$.

As shown in Appendix C, the action of $\mathsf{D}$ on the local operators $X_v$ and $Z_vZ_{v'}$ is identical to how the gauging of the $\mathbb{Z}_2^{(0)} \times \mathbb{Z}_2^{(1)}$ symmetry maps the local operators. Furthermore, the Hamiltonian (2.5) is self-dual under gauging the $\mathbb{Z}_2^{(0)} \times \mathbb{Z}_2^{(1)}$ symmetry. Hence $\mathsf{D}$ is a non-invertible duality symmetry associated with gauging (see Appendix A).

**Non-invertible symmetry in more general models**  The reader is welcome to further set $h' = J' = 0$ in (2.5). This will not affect the symmetries of interest to us. This special Hamiltonian is a decoupled model Ising $\times \frac{\text{Ising}}{\mathbb{Z}_2}$, with $\mathsf{D}$ exchanging the two sectors.[12] More generally, one can replace the Ising model by any lattice model $\mathcal{Q}$ with a $\mathbb{Z}_2^{(0)}$ symmetry, and consider the tensor product model $\mathcal{Q} \times \frac{\mathcal{Q}}{\mathbb{Z}_2}$. The latter Hamiltonian is invariant under a non-invertible operator $\mathsf{D}$ which swaps $\mathcal{Q}$ with $\frac{\mathcal{Q}}{\mathbb{Z}_2}$. One can further add interactions to couple the two sectors. See Appendix A for the continuum counterpart.

---

[12]The gauged Ising model $\frac{\text{Ising}}{\mathbb{Z}_2}$ is the lattice $\mathbb{Z}_2$ gauge theory. See Appendix C for more details on lattice gauging.

## 2.2 Non-invertible swap operator

Here we derive an explicit expression for the non-invertible operator $\mathsf{D}$. To proceed, we start with a different model, where there are two qubits on every site $v$, acted by the Pauli operators $(X_v, Z_v)$ and $(\tilde{X}_v, \tilde{Z}_v)$, respectively. The Hamiltonian is a 2+1d generalization of the 1+1d Ashkin-Teller model:

$$
\begin{aligned}
H_{\text{AT}} = & - h \sum_v X_v - J \sum_{\langle v,v' \rangle} Z_v Z_{v'} - \tilde{h} \sum_v \tilde{X}_v - \tilde{J} \sum_{\langle v,v' \rangle} \tilde{Z}_v \tilde{Z}_{v'} \\
& - h' \sum_v X_v \tilde{X}_v - J' \sum_{\langle v,v' \rangle} Z_v \tilde{Z}_v Z_{v'} \tilde{Z}_{v'} \,.
\end{aligned}
\tag{2.9}
$$

It has an on-site $\mathbb{Z}_2^{(0)} \times \tilde{\mathbb{Z}}_2^{(0)}$ symmetry generated by $U = \prod_v X_v, \tilde{U} = \prod_v \tilde{X}_v$. Restricting to $h = \tilde{h}, J = \tilde{J}$, which we assume throughout this section, there is an additional swap $\mathbb{Z}_{2,\text{swap}}^{(0)}$ symmetry that exchanges $(X_v, Z_v)$ with $(\tilde{X}_v, \tilde{Z}_v)$. It exchanges the two Ising models in the first line and leaves the terms in the second line invariant. The invertible swap operator is

$$
S = \prod_v S_v \,, \quad S_v = \frac{1}{2} \left( 1 + X_v \tilde{X}_v + Z_v \tilde{Z}_v - Z_v \tilde{Z}_v X_v \tilde{X}_v \right) \,.
\tag{2.10}
$$

The symmetry operators obey the algebra

$$
U^2 = \tilde{U}^2 = S^2 = 1 \,, \quad US = S\tilde{U} \,, \quad SU = \tilde{U}S \,, \quad U\tilde{U} = \tilde{U}U \,.
\tag{2.11}
$$

and together they form a $D_8$ symmetry, the dihedral group with 8 elements.[13]

As we will discuss momentarily, gauging the $\tilde{\mathbb{Z}}_2^{(0)}$ symmetry $\tilde{U}$ of $H_{\text{AT}}$ yields the original model (2.5). What happens to the swap symmetry $S$ under this gauging? We need to first understand what happens to $\tilde{X}_v$ and $\tilde{Z}_v$ under gauging.

When gauging the on-site symmetry $\tilde{U}$, we first introduce a $\mathbb{Z}_2$ gauge field on every link. Second, we impose the Gauss law and couple the gauge fields to the Hamiltonian minimally. We rewrite the gauged Hamiltonian in terms of a new set of gauge-invariant operators $(\sigma_\ell^x, \sigma_\ell^z)$ on the links. We review the gauging of a $\mathbb{Z}_2^{(0)}$ symmetry in detail in Appendix C.1.[14] The final result is that we replace the $\tilde{\mathbb{Z}}_2^{(0)}$-even local operators $\tilde{X}_v$ and $\tilde{Z}_v \tilde{Z}_{v'}$ by

$$
\tilde{X}_v \rightsquigarrow \prod_{\ell \ni v} \sigma_\ell^x \,, \quad \tilde{Z}_v \tilde{Z}_{v'} \rightsquigarrow \sigma_{\langle v,v' \rangle}^z \,.
\tag{2.12}
$$

---

[13]The $D_8$ group has an order 4 generator $a$ and an order 2 generator $b$. They obey the relation $a^4 = b^2 = 1, bab = a^3$. In terms of $U, \tilde{U}, S$, they are $(a, b) = (US, U)$.

[14]To apply the discussion of Appendix C.1 to here, we need to replace $X_v, Z_v$ there by $\tilde{X}_v, \tilde{Z}_v$.

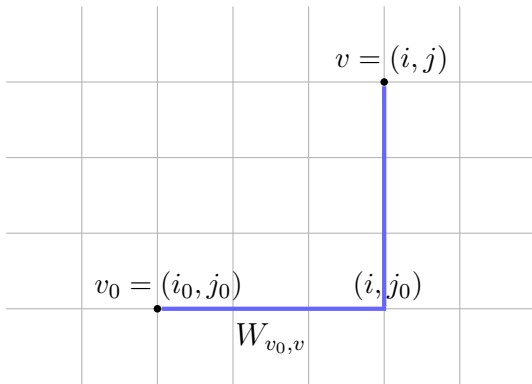

Figure 1: The Wilson line operator $W_{v_0,v}$ is a product of $\sigma_\ell^z$ along the blue curve from $v_0 = (i_0, j_0)$ to $v = (i, j)$.

In addition, we can add a flux term $\prod_{\ell \in f} \sigma_\ell^z$ to the gauged Hamiltonian. These steps then lead to the Hamiltonian (2.5). The gauged Hamiltonian has a dual 1-form symmetry $\eta$ in (2.2). The operators $X_v, Z_v$ are unaffected under gauging $\tilde{\mathbb{Z}}_2^{(0)}$, and $U$ is still a symmetry in the gauged model.

However, $S$ contains factors of $\tilde{Z}_v$, which is $\tilde{\mathbb{Z}}_2^{(0)}$-odd. After gauging, $\tilde{Z}_v$ is no longer a local operator. Intuitively, it becomes a (gauge-invariant) Wilson line operator denoted as $W_{v_0,v}$. More specifically, we pick an arbitrary reference point $v_0$, and multiply $\sigma_\ell^z$ from $v_0$ to $v$ along a curve. For now, we choose the curve to go from $v_0 = (i_0, j_0)$, to $(i, j_0)$, and then go upwards until it reaches $v = (i, j)$, as illustrated in Figure 1, where the pair $(i, j)$ denotes the $x, y$ coordinates of a vertex $v$. (Later, we will show that the final non-invertible operator is independent of the choice of $v_0$ and the above curve.) Explicitly, the Wilson operator is

$$W_{v_0,v} = \prod_{i'=i_0}^{i-1} \sigma_{i'+\frac{1}{2},j_0}^z \prod_{j'=j_0}^{j-1} \sigma_{i,j'+\frac{1}{2}}^z , \tag{2.13}$$

where $\sigma_{i+\frac{1}{2},j}^z$ denotes the link variable $\sigma_\ell^z$ at the link $\ell = (i + \frac{1}{2}, j)$.[15] For $v = v_0$, $W_{v_0,v_0} = 1$ is the identity operator.

To find out the fate of the swap symmetry under gauging, we proceed naively by substituting the Pauli operators $\tilde{X}_v$ and $\tilde{Z}_v$ in $S_v$ with $\prod_{\ell \ni v} \sigma_\ell^x$ and $W_{v_0,v}$, respectively. We denote the resulting operator as $\mathsf{S}_v$ with a different font:

$$\mathsf{S}_v = \frac{1}{2} \left[ 1 + G_v + Z_v W_{v_0,v} \left( 1 - G_v \right) \right] , \tag{2.14}$$

---

[15]This is not to be confused with the notation $\sigma_{\langle v,v' \rangle}^z = \sigma_\ell^z$ where $v$ and $v'$ denote the two endpoints of a link $\ell = \langle v, v' \rangle$

where

$$G_v = X_v \prod_{\ell \ni v} \sigma_\ell^x \tag{2.15}$$

and $v_0$ is an arbitrarily chosen reference point. Note that $\mathsf{S}_v$ is invertible for $v \neq v_0$, but non-invertible for $v = v_0$. It is not a local operator because of the Wilson line.

Similar to the invertible swap operator $S = \prod_v S_v$, it is tempting to take a product of $\mathsf{S}_v$ for all $v$ to find a symmetry operator of the gauged Hamiltonian (2.5). However, there are two issues with this:

1. While $\mathsf{S}_v$ and $\mathsf{S}_{v'}$ commute for $v, v' \neq v_0$, the operator $\mathsf{S}_{v_0}$ doesn't commute with $\mathsf{S}_v$ for $v \neq v_0$. Therefore, the ordering of $\mathsf{S}_v$'s needs to be specified.

2. $\mathsf{S}_v$ depends on a reference point $v_0$ and a choice of a curve in Figure 1. However, we expect the symmetry operator $\mathsf{D}$ to be translationally invariant and independent of the choice of the curve.

To proceed, we define the operator

$$\mathsf{S} \equiv \left( \prod_{v \neq v_0} \mathsf{S}_v \right) \mathsf{S}_{v_0} , \tag{2.16}$$

where we have placed $\mathsf{S}_{v_0}$ at the rightmost. (The ordering in the product $\prod_{v \neq v_0}$ does not matter because $\mathsf{S}_v \mathsf{S}_{v'} = \mathsf{S}_{v'} \mathsf{S}_v$ for $v, v' \neq v_0$.) Note that since $\mathsf{S}_{v_0}$ is non-invertible, so is $\mathsf{S}$.

In Appendix E.1, we show that $\mathsf{S}$ acts on most of the $X_v$'s in the desired way (2.7):

$$\mathsf{S} X_v = \left( \prod_{\ell \ni v} \sigma_\ell^x \right) \mathsf{S} , \quad X_v \mathsf{S} = \mathsf{S} \left( \prod_{\ell \ni v} \sigma_\ell^x \right) , \quad \forall\, v \neq v_0 . \tag{2.17}$$

However, for $v = v_0$, $\mathsf{S}$ does not map $X_v$ to $\prod_{\ell \ni v} \sigma_\ell^x$:

$$\mathsf{S} X_{v_0} = \left( \prod_{\ell \ni v_0} \sigma_\ell^x \right) \mathsf{S} U , \quad X_{v_0} \mathsf{S} = \mathsf{S} \left( \prod_{\ell \ni v_0} \sigma_\ell^x \right) U , \tag{2.18}$$

where $U = \prod_v X_v$. This instructs us to multiply (twice) the projection operator $\mathsf{C}_U = 1 + U$ to the right of $\mathsf{S}$ so that the resulting operator $\mathsf{S}\mathsf{C}_U$ maps $X_v$ to $\prod_{\ell \ni v} \sigma_\ell^x$ and vice versa for all $v$. That is,

$$(\mathsf{S}\mathsf{C}_U) X_v = \left( \prod_{\ell \ni v} \sigma_\ell^x \right) (\mathsf{S}\mathsf{C}_U) , \quad X_v (\mathsf{S}\mathsf{C}_U) = (\mathsf{S}\mathsf{C}_U) \left( \prod_{\ell \ni v} \sigma_\ell^x \right) , \quad \forall\, v . \tag{2.19}$$

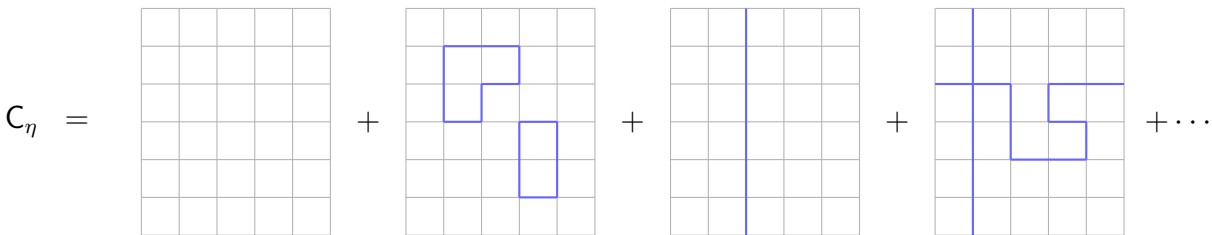

Figure 2: The condensation operator $\mathsf{C}_\eta$ is defined as a sum of the $\mathbb{Z}_2^{(1)}$ 1-form symmetry operators $\eta(\gamma)$ over all possible loops $\gamma$ on a periodic square lattice. The sum includes loops in the trivial homology class (for instance, the first two terms on the right-hand side) as well as loops in non-trivial homology classes (for instance, the third and fourth terms on the right-hand side).

Next, we need to check if $\mathsf{SC}_U$ acts correctly on the operator $Z_v Z_{v'}$ as in (2.7). However, we find (see Appendix E.1)

$$(\mathsf{SC}_U)Z_v Z_{v'} = W_{v_0,v}W_{v_0,v'}(\mathsf{SC}_U)\,, \quad Z_v Z_{v'}(\mathsf{SC}_U) = (\mathsf{SC}_U)W_{v_0,v}W_{v_0,v'}\,, \quad \forall\, v, v'\,, \qquad (2.20)$$

which, for neighboring vertices $v, v'$, does not generally give $\sigma^z_{\langle v,v'\rangle}$. To fix this, we define the *condensation operator* as

$$\mathsf{C}_\eta = \frac{1}{2^A}\sum_\gamma \eta(\gamma)\,, \qquad (2.21)$$

where $A = L_x L_y$ is the total number of faces (or sites), and the sum is over every loop $\gamma$ on a periodic square lattice (including the empty loop). See Figure 2.

We discuss the condensation operator in more detail in Section 2.3. For now, one defining property that we need is that it absorbs the 1-form symmetry operator,

$$\mathsf{C}_\eta \eta(\gamma) = \eta(\gamma)\mathsf{C}_\eta = \mathsf{C}_\eta\,, \quad \forall\, \gamma\,. \qquad (2.22)$$

In particular, when $v$ and $v'$ are two neighboring sites, we have

$$W_{v_0,v}W_{v_0,v'}\mathsf{C}_\eta = \sigma^z_{\langle v,v'\rangle}\mathsf{C}_\eta\,, \qquad (2.23)$$

since $W_{v_0,v}W_{v_0,v'}$ and $\sigma^z_{\langle v,v'\rangle}$ always differ by a one-form symmetry loop $\eta(\gamma)$ where $\gamma$ could either be homologically trivial or non-trivial. See Figure 3. Combined with (2.20), this implies that the product $\mathsf{SC}_U\mathsf{C}_\eta$ acts on $Z_v Z_{v'}$ in the desired way (2.7).[16] Moreover, since $\mathsf{C}_\eta$ acts only on the link variables, it doesn't affect the action on $X_v$ that we have already fixed.

To summarize, we multiply the naive operator $\mathsf{S}$ by $\mathsf{C}_U$ and $\mathsf{C}_\eta$ to fix the action on $X_v$ and

---

[16]Since $\mathsf{C}_\eta$ commutes with $\mathsf{SC}_U$, we can multiply it on either side of $\mathsf{SC}_U$.

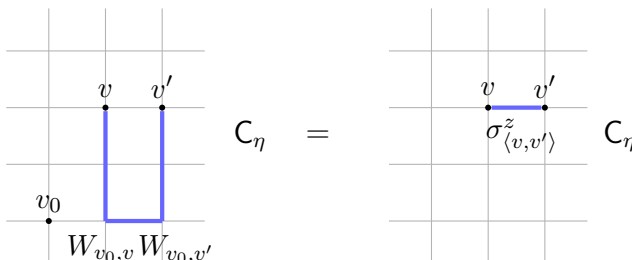

Figure 3: Upon multiplying by the condensation operator $\mathsf{C}_\eta$, we can replace $W_{v_0,v}W_{v_0,v'}$ with adjacent $v,v'$ by the link variable $\sigma^z_{\langle v,v'\rangle}$. (Note that the two $\sigma^z_\ell$ operators from the two Wilson lines cancel with each other on the link immediately on the right of $v_0$.)

$Z_v Z_{v'}$, respectively. Putting everything together, we define the final non-invertible operator as

$$\mathsf{D} = \frac{1}{2}\mathsf{S}\mathsf{C}_U\mathsf{C}_\eta = \frac{1}{2}\mathsf{S}\mathsf{C}\,, \tag{2.24}$$

where we have defined $\mathsf{C} = \mathsf{C}_U\mathsf{C}_\eta$. This operator acts on the local operators as in (2.7). Furthermore, we show that it is hermitian, i.e., $\mathsf{D} = \mathsf{D}^\dagger$, in Appendix E.3. The normalization factor $1/2$ is chosen for later convenience so that it agrees with the continuum expression.

Finally, even though $\mathsf{S}$ depends on $v_0$, in Appendix E.2 we show that $\mathsf{D}$ is independent of $v_0$. It is also independent of the choice of the curve in $W_{v_0,v}$ because of the condensation operator $\mathsf{C}_\eta$. We have thus established a non-invertible symmetry operator of (2.5).

## 2.3 Condensation operator and the toric code states

In this section we further discuss the condensation operator $\mathsf{C}_\eta$ (2.21) associated with the 1-form symmetry $\eta(\gamma) = \prod_{\ell\in\gamma}\sigma^z_\ell$. The discussion here applies to any Hamiltonian with this 1-form symmetry, which includes the toric code as a special case.

The condensation operator $\mathsf{C}_\eta$ admits an alternative presentation,

$$\mathsf{C}_\eta = \frac{1}{2}(1+\eta_x)(1+\eta_y)\prod_f \frac{1}{2}(1+B_f)\,, \tag{2.25}$$

where $B_f = \prod_{\ell\in f}\sigma^z_\ell$ and $\eta_i = \prod_{\ell\in\gamma_i}\sigma^z_\ell$ with $\gamma_i$ a non-contractible cycle wrapping around the $i$-direction.[17]

If we expand all the products on the right-hand side of (2.25), we recover the previous expression (2.21). To see this, first note that one of the terms in the product $\prod_f \frac{1}{2}(1+B_f)$

---

[17]Since $\eta_x, \eta_y$ are always multiplied by $\prod_f \frac{1}{2}(1+B_f)$, the choice of the representative cycle does not matter.

is redundant. That is, $\prod_f \frac{1}{2}(1 + B_f) = \prod_{f \neq f_0} \frac{1}{2}(1 + B_f)$ for an arbitrarily chosen face $f_0$ because $\prod_f B_f = 1$. Furthermore, we have

$$\prod_{f \neq f_0} \frac{1}{2}(1 + B_f) = \frac{1}{2^{A-1}} \sum_{\text{trivial } \gamma} \eta(\gamma), \qquad (2.26)$$

where the sum on the right-hand side is restricted to be over closed loops $\gamma$ which are homologically trivial. Plugging this back into (2.25), we get

$$\mathsf{C}_\eta = \frac{1}{2}(1 + \eta_x)(1 + \eta_y) \times \frac{1}{2^{A-1}} \left( \sum_{\text{trivial } \gamma} \eta(\gamma) \right) = \frac{1}{2^A} \sum_\gamma \eta(\gamma), \qquad (2.27)$$

where now the sum runs over all possible loops $\gamma$, both homologically trivial and nontrivial ones. Therefore, we see that (2.25) is equivalent to the expression (2.21).

If we impose the Gauss law $B_f = 1$ strictly by sending $g \to \infty$, the 1-form symmetry operator $\eta(\gamma)$ becomes topological. The condensation operator (2.25) reduces to

$$\mathsf{C}_\eta = \frac{1}{2}(1 + \eta_x)(1 + \eta_y) \quad \text{on} \quad \widetilde{\mathcal{H}}. \qquad (2.28)$$

This agrees with the definition of the condensation operator $\mathcal{C}_\eta(\Sigma)$ in 2+1d continuum QFTs [65, 66], defined on a two-manifold $\Sigma$:[18]

$$\mathcal{C}_\eta(\Sigma) = \frac{1}{|H^0(\Sigma, \mathbb{Z}_2)|} \sum_{\gamma \in H_1(\Sigma, \mathbb{Z}_2)} \eta(\gamma), \qquad (2.29)$$

where we also use $\eta(\gamma)$ to denote the $\mathbb{Z}_2^{(1)}$ symmetry operator in the continuum. Indeed, when $\Sigma$ is chosen to be a two-torus, the above two expressions become identical.

For the ordinary $\mathbb{Z}_2^{(0)}$ symmetry generated by $U$, one can trivially view twice the projection operator $\mathsf{C}_U = 1 + U$ as its condensation operator. It is convenient to define the total condensation operator for the $\mathbb{Z}_2^{(0)} \times \mathbb{Z}_2^{(1)}$ symmetry as $\mathsf{C} = \mathsf{C}_U \mathsf{C}_\eta$. Clearly, we have

$$\mathsf{C}U = U\mathsf{C} = \mathsf{C}, \quad \mathsf{C}\eta(\gamma) = \eta(\gamma)\mathsf{C} = \mathsf{C}. \qquad (2.30)$$

---

[18]The normalization of the condensation operator here differs from the one in [65] by an Euler counterterm. More generally, in the continuum, the condensation operator $\mathcal{C}_\eta(\Sigma)$ can be defined if the $\mathbb{Z}_2$ topological line $\eta$ is a boson or a fermion.

**Toric code ground states**  One distinguished feature of the condensation operator is that it creates a toric code ground state from a product state.[19]  (For the following discussion we focus on the qubits on the links.)  On a spatial two-torus, the 2+1d toric code (whose Hamiltonian is the first two terms of (1.2)) has four exactly degenerate ground states in finite volume. We can choose a basis of these states as follows [57]:

$$|\xi\rangle = 2^{\frac{A-1}{2}} \eta_x^{\xi_x} \eta_y^{\xi_y} \prod_f \frac{1 + B_f}{2} |+\cdots+\rangle_{\text{link}}, \tag{2.31}$$

where $\xi = (\xi_x, \xi_y)$ with $\xi_i = 0, 1$ defined modulo 2.  In this basis, the 1-form symmetry operators permute the four states:

$$\eta_x^{\xi_x'} \eta_y^{\xi_y'} |\xi\rangle = |\xi + \xi'\rangle. \tag{2.32}$$

Alternatively, we can choose a different basis:

$$|\zeta\rangle = \frac{1}{2} \sum_{\xi \in \{0,1\}^2} (-1)^{\zeta_x \xi_x + \zeta_y \xi_y} |\xi\rangle, \tag{2.33}$$

which diagonalizes the 1-form symmetry operators:

$$\eta_i |\zeta\rangle = (-1)^{\zeta_i} |\zeta\rangle. \tag{2.34}$$

In particular, the ground state $|\zeta = 0\rangle$, which has $\eta_x = \eta_y = +1$, is created by the condensation operator from the product state:[20]

$$|\zeta = 0\rangle = 2^{\frac{A-1}{2}} \mathsf{C}_\eta |+\cdots+\rangle_{\text{link}}. \tag{2.35}$$

## 2.4  Operator algebra

We derive the operator algebra between the duality operator $\mathsf{D}$ and the $\mathbb{Z}_2^{(0)} \times \mathbb{Z}_2^{(1)}$ symmetry in Appendix E.4:

$$\mathsf{D}^2 = (1 + U) \times \frac{1}{2^A} \sum_\gamma \eta(\gamma),$$

$$\mathsf{D}U = U\mathsf{D} = \mathsf{D}\eta(\gamma) = \eta(\gamma)\mathsf{D} = \mathsf{D}. \tag{2.36}$$

---

[19]We thank Weiguang Cao, Pranay Gorantla, and Sahand Seifnashri for discussions on this point.

[20]The toric Hamiltonian has another 1-form symmetry, whose condensation operator creates the $|\xi = 0\rangle$ state. See also [67] for the sequential quantum circuit that creates the toric code ground state.

In terms of the condensation operator $\mathsf{C}$, the algebra becomes

$$\mathsf{D}^2 = \mathsf{C}, \quad \mathsf{C}^2 = 4\mathsf{C}, \quad \mathsf{DC} = \mathsf{CD} = 4\mathsf{D}. \tag{2.37}$$

Unlike the non-invertible Kramers-Wannier lattice operator in 1+1d [15, 16] (see also [40]) and the Wegner duality operator in 3+1d lattice $\mathbb{Z}_2$ gauge theory [68], here the non-invertible symmetry does not mix with the lattice translation.

The continuum counterpart of these lattice symmetries forms a 2-Rep$((\mathbb{Z}_2^{(1)} \times \mathbb{Z}_2^{(1)}) \rtimes \mathbb{Z}_2^{(0)})$ fusion 2-category, which was discussed in [60, 61]. It is a 2-category of 2-representations of the 2-group $(\mathbb{Z}_2^{(1)} \times \mathbb{Z}_2^{(1)}) \rtimes \mathbb{Z}_2^{(0)}$, where the $\mathbb{Z}_2^{(0)}$ 0-form symmetry exchanges two 1-form symmetries, and the Postnikov class is trivial.[21] This fusion 2-category in 2+1d is a natural generalization of the Rep$(D_8) = $ Rep$(\mathbb{Z}_2^2 \rtimes \mathbb{Z}_2)$ fusion category in 1+1d [28].[22] See also [69–71] for related discussions of fusion 2-categories in continuum field theory.

# 3   Anomaly involving the lattice 1-form symmetry

In this section, we discuss an exact lattice 1-form symmetry with a mixed 't Hooft anomaly with other ordinary symmetries. We then study its implication on the phase diagram.

## 3.1   Hamiltonian and symmetry operators

We continue to analyze the Hamiltonian (1.3) for the lattice $\mathbb{Z}_2$ gauge theory coupled to Ising matter. In Section 2, we discussed the case $h = \tilde{h}, J = \tilde{J}$ where the model has a (non-anomalous) non-invertible symmetry. In this section, we consider another limit $h = h', J' = \tilde{J}$, where the model enjoys an interesting mixed anomaly.

The Hamiltonian is

$$\begin{aligned}
H_{\mathrm{anom}} = &- h \sum_v X_v - J \sum_{\langle v,v' \rangle} Z_v Z_{v'} - \tilde{h} \sum_v \prod_{\ell \ni v} \sigma_\ell^x - g \sum_f \prod_{\ell \in f} \sigma_\ell^z - \tilde{J} \sum_\ell \sigma_\ell^z \\
&- h \sum_v X_v \prod_{\ell \ni v} \sigma_\ell^x - \tilde{J} \sum_{\langle v,v' \rangle} Z_v \sigma_{\langle v,v' \rangle}^z Z_{v'}.
\end{aligned} \tag{3.1}$$

It has a $\mathbb{Z}_2^{(0)} \times \mathbb{Z}_2^{(1)}$ symmetry generated by $U$ and $\eta(\gamma)$ (which also exist for the more general Hamiltonian (1.3)). Note that this Hamiltonian includes the toric code Hamiltonian as a

---

[21] Here the $\mathbb{Z}_2^{(1)} \times \mathbb{Z}_2^{(1)}$ 1-form symmetry is the dual symmetry after gauging $\mathbb{Z}_2^{(0)} \times \tilde{\mathbb{Z}}_2^{(0)}$ generated by $U$ and $\tilde{U}$ in $D_8$.

[22] Another natural generalization of Rep$(D_8)$ to 2+1d is the fusion 2-category 2-Rep$(D_8)$.

special case when $h = J = \tilde{J} = 0$, and our discussions of the 1-form $\mathbb{Z}_2^{(1)}$ symmetry itself, as well as its condensation operator in Section 2.3, also apply to the toric code.

In addition, there is another invertible, ordinary symmetry $\mathbb{Z}_{2,\text{swap}}^{(0)}$ generated by

$$V = \prod_v V_v \,, \quad V_v = \frac{1}{2}\left( (1 + Z_v) + (1 - Z_v) \prod_{\ell \ni v} \sigma_\ell^x \right) . \tag{3.2}$$

It exchanges the terms in the Hamiltonian as

$$V X_v V^{-1} = X_v \prod_{\ell \ni v} \sigma_\ell^x \,, \quad V \sigma_{\langle v,v' \rangle}^z V^{-1} = Z_v \sigma_{\langle v,v' \rangle}^z Z_{v'} \,, \quad V \left( \prod_{\ell \in f} \sigma_\ell^z \right) V^{-1} = \prod_{\ell \in f} \sigma_\ell^z \,. \tag{3.3}$$

The operators $U, \eta(\gamma), V$ mutually commute, hence they generate a $\mathbb{Z}_2^{(0)} \times \mathbb{Z}_2^{(1)} \times \mathbb{Z}_{2,\text{swap}}^{(0)}$ symmetry.

Each of $U, \eta(\gamma), V$ is free of 't Hooft anomaly and can be gauged separately. In particular, we discuss the gauging of $U$ and $\eta$ in Appendix C. However, in Section 3.3, we will show that there is a mixed anomaly among them.

The reader is welcome to further set $\tilde{h} = J = 0$ in (3.1). This will not affect the symmmetries and anomalies of interest to us. This special Hamiltonian is $(\text{Ising} \times \text{Ising})/\mathbb{Z}_2$, where $\mathbb{Z}_2$ is the diagonal spin-flip symmetry. The symmetry $V$ exchanges the two Ising factors. See Appendix C.3 for details. More generally, one can replace the Ising model by any lattice model $\mathcal{Q}$ with a $\mathbb{Z}_2^{(0)}$ symmetry, and consider $(\mathcal{Q} \times \mathcal{Q})/\mathbb{Z}_2$. One can further add interactions that couple the two $\mathcal{Q}$'s before the diagonal gauging. See Appendix B for the continuum counterpart.

## 3.2 Defects and their equivalence classes

Given the symmetry operators $U, V$, and $\eta(\gamma)$, we consider the corresponding symmetry defects [72, 73]. The defect is given by a local modification of the Hamiltonian (3.1) along a 1-dimensional (0-dimensional) locus in space for a 0-form (1-form) symmetry.[23] When the 0-form symmetry defect is inserted along a nontrivial cycle, it implements a twisted boundary condition.

---

[23]In spacetime, these defects extend in the time direction as well. Therefore, the defect for a 0-form (1-form) symmetry is 2-dimensional (1-dimensional) in spacetime, matching the picture in the continuum [54].

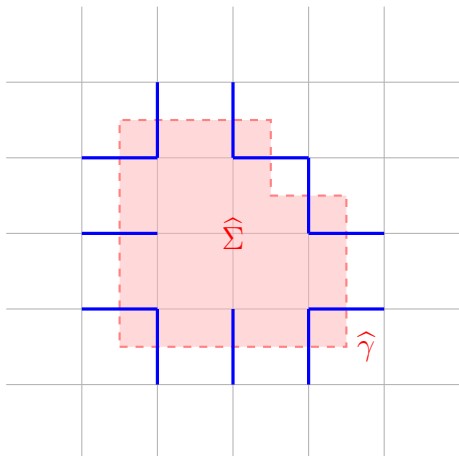

Figure 4: Configuration of the membrane operator $U(\widehat{\Sigma})$. Here $\widehat{\Sigma}$ is the red region, and its boundary (represented by the red dashed line) is the curve $\widehat{\gamma}$ on the dual lattice. Those links $\langle v, v' \rangle$ having a non-trivial intersection with $\widehat{\gamma}$, i.e., $(\langle v, v' \rangle, \widehat{\gamma}) = 1$, are shown in blue.

### 3.2.1 Defect for $\mathbb{Z}_2^{(0)}$

Consider first the $\mathbb{Z}_2^{(0)}$ symmetry generated by the $U$ operator. We define a *membrane operator* by cutting open the symmetry operator $U$,

$$U(\widehat{\Sigma}) \equiv \prod_{v \in \widehat{\Sigma}} X_v \,. \tag{3.4}$$

Here, $\widehat{\Sigma}$ denotes a dual 2-chain composed of dual plaquettes and the product on the right-hand side is over the vertices $v$ which intersect nontrivially with $\widehat{\Sigma}$. See Figure 4. When $\widehat{\Sigma}$ covers the entire space (which is a two-dimensional torus), the membrane operator $U(\widehat{\Sigma})$ simply reduces to the original symmetry operator $U$. The choice of a membrane operator $U(\widehat{\Sigma})$ is not unique, and other choices will be discussed later in Section 3.2.4.

The defect Hamiltonian for the symmetry operator $U$ is obtained by conjugating the original Hamiltonian with the membrane operator:

$$
\begin{aligned}
H_{\text{anom},U}(\widehat{\gamma}) &\equiv U(\widehat{\Sigma}) H_{\text{anom}} U(\widehat{\Sigma})^{-1} \\
&= -h \sum_v X_v - J \sum_{\langle v,v' \rangle} (-1)^{(\widehat{\gamma}, \langle v,v' \rangle)} Z_v Z_{v'} - \tilde{h} \sum_v \prod_{\ell \ni v} \sigma_\ell^x - g \sum_f \prod_{\ell \in f} \sigma_\ell^z \\
&\quad - \tilde{J} \sum_\ell \sigma_\ell^z - h \sum_v X_v \prod_{\ell \ni v} \sigma_\ell^x - \tilde{J} \sum_{\langle v,v' \rangle} (-1)^{(\widehat{\gamma}, \langle v,v' \rangle)} Z_v \sigma_{\langle v,v' \rangle}^z Z_{v'} \,.
\end{aligned}
\tag{3.5}
$$

Here, $\widehat{\gamma} = \partial \widehat{\Sigma}$ is the dual 1-cycle which is the boundary locus of the membrane operator,

and $(\widehat{\gamma}, \langle v, v' \rangle)$ is the intersection number between the link $\langle v, v' \rangle$ and the line $\widehat{\gamma}$. The defect Hamiltonian $H_{\mathrm{anom},U}(\widehat{\gamma})$ is identical to the original Hamiltonian $H_{\mathrm{anom}}$ except along the 1-dimensional locus $\widehat{\gamma}$. Specifically, the signs of the terms on the links $\langle v, v' \rangle$ satisfying $(\widehat{\gamma}, \langle v, v' \rangle) = 1$ (i.e., the blue links in Figure 4) are flipped.

More generally, the defect Hamiltonian is defined with the same expression for an arbitrary dual 1-cycle $\widehat{\gamma}$ which is not necessarily a boundary of a dual 2-chain $\widehat{\Sigma}$. When $\widehat{\gamma}$ is homologically nontrivial, the defect Hamiltonian $H_{\mathrm{anom},U}$ is not related to the original Hamiltonian $H_{\mathrm{anom}}$ by a unitary operator, and the defect represents a twisted boundary condition for the $\mathbb{Z}_2^{(0)}$ symmetry generated by $U$.

The symmetry defect is *topological*, in the sence that the energy spectrum of the defect Hamiltonian $H_{\mathrm{anom},U}(\widehat{\gamma})$ depends only on the homology class of the defect locus $\widehat{\gamma}$ but not on its detailed shape. In particular, using the unitary membrane operator (3.4), we can freely deform the defect locus:

$$U(\widehat{\Sigma}')H_{\mathrm{anom},U}(\widehat{\gamma})U(\widehat{\Sigma}')^{-1} = H_{\mathrm{anom},U}(\widehat{\gamma} + \partial\widehat{\Sigma}') . \tag{3.6}$$

Here, $\widehat{\gamma} + \partial\widehat{\Sigma}'$ is the sum of ($\mathbb{Z}_2$-valued) 1-cycles. The membrane operator plays a similar role as the movement operator for ordinary symmetry defects in 1+1d [73].

### 3.2.2 Defect for $\mathbb{Z}_{2,\mathbf{swap}}^{(0)}$

The defect Hamiltonian for the $\mathbb{Z}_{2,\mathrm{swap}}^{(0)}$ symmetry operator $V$ can be similarly obtained. We first define a membrane operator (again the choice of a membrane operator is not unique),

$$V(\widehat{\Sigma}) = \prod_{v \in \widehat{\Sigma}} \left[ \frac{1}{2} \left( (1 + Z_v) + (1 - Z_v) \prod_{\ell \ni v} \sigma_\ell^x \right) \right] . \tag{3.7}$$

Conjugating the Hamiltonian with the membrane operator $V(\widehat{\Sigma})$, we can read off the defect Hamiltonian:

$$H_{\mathrm{anom},V}(\widehat{\gamma}) = -h \sum_v X_v - J \sum_{\langle v,v' \rangle} Z_v Z_{v'} - \tilde{h} \sum_v \prod_{\ell \ni v} \sigma_\ell^x - \tilde{J} \sum_{(\widehat{\gamma}, \langle v,v' \rangle)=0} \left( Z_v \sigma_{\langle v,v' \rangle}^z Z_{v'} + \sigma_{\langle v,v' \rangle}^z \right)$$
$$- \tilde{J} \sum_{(\widehat{\gamma}, \langle v,v' \rangle)=1} \left( Z_v \sigma_{\langle v,v' \rangle}^z + \sigma_{\langle v,v' \rangle}^z Z_{v'} \right) - g \sum_f \prod_{\ell \in f} \sigma_\ell^z - h \sum_v X_v \prod_{\ell \ni v} \sigma_\ell^x . \tag{3.8}$$

The expression (3.8) for the defect Hamiltonian is valid for an arbitrary dual 1-cycle $\widehat{\gamma}$. The defect is again topological similarly to (3.6), with $U$ replaced by $V$.

### 3.2.3 Defect for $\mathbb{Z}_2^{(1)}$

The defect Hamiltonian for the $\mathbb{Z}_2^{(1)}$ 1-form symmetry is given by a local modification of the original Hamiltonian at a single lattice site. We define an *interval operator* along an interval $\mathcal{P}$ by cutting open the symmetry operator $\eta$

$$\eta(\mathcal{P}) = \prod_{\ell \in \mathcal{P}} \sigma_\ell^z \, . \tag{3.9}$$

The boundary of the interval $\partial \mathcal{P}$ consists of two sites $v_0, v_1$. Conjugating the Hamiltonian by the operator $\eta(\mathcal{P})$ creates a pair of 1-form symmetry defects located at sites $v_0$ and $v_1$, respectively. From this, we can read off the expression for the defect Hamiltonian for a single defect located at the vertex $v_0$:

$$
\begin{aligned}
H_{\mathrm{anom},\eta}(v_0) = &- h \sum_v X_v - J \sum_{\langle v,v' \rangle} Z_v Z_{v'} - \tilde{h} \sum_v (-1)^{\delta_{v_0,v}} \prod_{\ell \ni v} \sigma_\ell^x - g \sum_f \prod_{\ell \in f} \sigma_\ell^z - \tilde{J} \sum_\ell \sigma_\ell^z \\
&- h \sum_v (-1)^{\delta_{v_0,v}} X_v \prod_{\ell \ni v} \sigma_\ell^x - \tilde{J} \sum_{\langle v,v' \rangle} Z_v \sigma_{\langle v,v' \rangle}^z Z_{v'} \, .
\end{aligned}
\tag{3.10}
$$

The $\eta$ defect is again topological similarly to (3.6), with $U$ replaced by $\eta$.

### 3.2.4 Equivalence class of defects

In deriving the defect Hamiltonians, we had to first cut open a symmetry operator and define an open operator, i.e., membrane operator for $U, V$, and interval operator for $\eta$. However, the choice of an open operator for a given symmetry operator is generally not unique. For instance, consider the symmetry operator $U = \prod_v X_v$. Instead of (3.4), we may alternatively cut open the $U$ operator by defining a different membrane operator:

$$U'(\widehat{\Sigma}) \equiv \prod_{v \in \widehat{\Sigma}} \left( X_v \prod_{\ell \ni v} \sigma_\ell^x \right) \, . \tag{3.11}$$

The new membrane operator $U'(\widehat{\Sigma})$ is the same as $U(\widehat{\Sigma})$ in the interior of $\widehat{\Sigma}$ since $\sigma_\ell^x$'s cancel, they only differ at the boundary $\partial \widehat{\Sigma}$.

Using (3.11), one obtains a different expression of the defect Hamiltonian,

$$
\begin{aligned}
H_{\mathrm{anom},U'}(\widehat{\gamma}) = &-h\sum_v X_v - J\sum_{\langle v,v'\rangle}(-1)^{(\widehat{\gamma},\langle v,v'\rangle)}Z_v Z_{v'} - \tilde{h}\sum_v \prod_{\ell\ni v}\sigma_\ell^x - g\sum_f \prod_{\ell\in f}\sigma_\ell^z \\
&-\tilde{J}\sum_\ell (-1)^{(\widehat{\gamma},\ell)}\sigma_\ell^z - h\sum_v X_v \prod_{\ell\ni v}\sigma_\ell^x - \tilde{J}\sum_{\langle v,v'\rangle} Z_v \sigma_{\langle v,v'\rangle}^z Z_{v'}\,.
\end{aligned}
\tag{3.12}
$$

Although $H_{\mathrm{anom},U'}(\widehat{\gamma})$ is different from the previous defect Hamiltonian $H_{\mathrm{anom},U}(\widehat{\gamma})$ in (3.5), the two differ only by a finite-depth unitary transformation which is localized along the defect locus $\widehat{\gamma}$,

$$
H_{\mathrm{anom},U'}(\widehat{\gamma}) = W(\widehat{\gamma})H_{\mathrm{anom},U}(\widehat{\gamma})W(\widehat{\gamma})^{-1}\,,
\tag{3.13}
$$

where

$$
W(\widehat{\gamma}) = \prod_{(\ell,\widehat{\gamma})=1}\sigma_\ell^x\,.
\tag{3.14}
$$

The same discussion applies to defect Hamiltonians for $V$ and $\eta$.

In general, we say that two topological defects are in the same *equivalence class* if the corresponding defect Hamiltonians are related by a finite-depth unitary transformation which acts only along the locus of defect. Different choices of an open operator generally lead to the same equivalence class of the topological defect.[24] Similar discussions can be found, for instance, in [74]. Later in Section 3.3, we will see that the 't Hooft anomaly depends only on the equivalence class of the defects.

## 3.3 Type III anomaly between 0- and 1-form symmetries

We now show that there exists a mixed 't Hooft anomaly in $\mathbb{Z}_2^{(0)}\times\mathbb{Z}_2^{(1)}\times\mathbb{Z}_{2,\mathrm{swap}}^{(0)}$. This 't Hooft anomaly is an obstruction to gauging these three symmetries at the same time. Anomalies of lattice higher-form symmetries have been studied in the past, for instance, in [62, 75, 76].

Our general strategy to detect the mixed anomaly follows [72, 28]. See also [77, 73] for the analogous discussion for LSM anomalies. We will show that in the presence of a defect for one of the three symmetries, the other two symmetry operators form a projective algebra. Appendix B discusses the corresponding anomaly inflow action in the continuum.

---

[24]From the continuum QFT point of view, the finite-depth unitary transformation localized along the defect locus can be thought of as an invertible topological interface between the two defects. In the language of (higher-)category theory, the existence of such an invertible interface means that the two defects are isomorphic [69].

$\mathbb{Z}_2^{(1)}$ **twist**  Consider first the defect Hamiltonian for the 1-form symmetry $H_{\mathrm{anom},\eta}(v_0)$ given in (3.10). In the presence of the 1-form symmetry defect, the $U = \prod_v X_v$ symmetry operator still commutes with the defect Hamiltonian $H_{\mathrm{anom},\eta}(v_0)$. However, the $\mathbb{Z}_{2,\mathrm{swap}}^{(0)}$ symmetry operator $V$ given in (3.2) no longer commutes with $H_{\mathrm{anom},\eta}(v_0)$. Instead, we modify the operator $V$ at the defect location and define

$$V(v_0) \equiv Z_{v_0} V \, , \tag{3.15}$$

so that $V(v_0) H_{\mathrm{anom},\eta}(v_0) = H_{\mathrm{anom},\eta}(v_0) V(v_0)$. However, this modified symmetry operator $V(v_0)$, together with $U$, now forms a projective $\mathbb{Z}_2^{(0)} \times \mathbb{Z}_{2,\mathrm{swap}}^{(0)}$ algebra,

$$U V(v_0) = -V(v_0) U \, . \tag{3.16}$$

Generally, one finds that it is not possible to realize the $\mathbb{Z}_2^{(0)} \times \mathbb{Z}_{2,\mathrm{swap}}^{(0)}$ symmetry linearly in the presence of the 1-form symmetry defect, hence signaling the existence of a mixed anomaly involving the three symmetries $\mathbb{Z}_2^{(0)} \times \mathbb{Z}_2^{(1)} \times \mathbb{Z}_{2,\mathrm{swap}}^{(0)}$.

$\mathbb{Z}_2^{(0)}$ **twist**  Alternatively, consider the defect Hamiltonian $H_{\mathrm{anom},U}(\widehat{\gamma})$ in (3.5). The 1-form symmetry operator $\eta(\gamma)$ still commutes $H_{\mathrm{anom},U}(\widehat{\gamma})$, but the $\mathbb{Z}_{2,\mathrm{swap}}^{(0)}$ symmetry operator needs to be modified along the locus $\widehat{\gamma}$:

$$V(\widehat{\gamma}) \equiv V \prod_{(\ell,\widehat{\gamma})=1} \sigma_\ell^x \, , \tag{3.17}$$

to ensure $V(\widehat{\gamma}) H_{\mathrm{anom},U}(\widehat{\gamma}) = H_{\mathrm{anom},U}(\widehat{\gamma}) V(\widehat{\gamma})$. Consequently, the $\mathbb{Z}_{2,\mathrm{swap}}^{(0)} \times \mathbb{Z}_2^{(1)}$ symmetry algebra is realized projectively in the presence of the $U$ defect,

$$V(\widehat{\gamma}) \eta(\gamma) = (-1)^{(\gamma,\widehat{\gamma})} \eta(\gamma) V(\widehat{\gamma}) \, . \tag{3.18}$$

Such a projective algebra in the presence of a defect again signals a mixed anomaly.

$\mathbb{Z}_{2,\mathrm{swap}}^{(0)}$ **twist**  Finally, for the defect Hamiltonian $H_V(\widehat{\gamma})$ given in (3.8), the 1-form symmetry operator $\eta(\gamma)$ still commutes with it, but the $U$ operator needs to be modified:

$$U(\widehat{\gamma}) \equiv U \prod_{(\ell,\widehat{\gamma})=1} \sigma_\ell^x \, , \tag{3.19}$$

to ensure $U(\widehat{\gamma})H_{\mathrm{anom},V}(\widehat{\gamma}) = H_{\mathrm{anom},V}(\widehat{\gamma})U(\widehat{\gamma})$. Consequently, $\mathbb{Z}_2^{(0)} \times \mathbb{Z}_2^{(1)}$ symmetry algebra is realized projectively:

$$U(\widehat{\gamma})\eta(\gamma) = (-1)^{(\gamma,\widehat{\gamma})}\eta(\gamma)U(\widehat{\gamma})\,, \tag{3.20}$$

indicating the mixed anomaly.

The projective algebras (3.16), (3.18) and (3.20) are not affected if we instead use the equivalent defect Hamiltonians given in (3.12) and its $V, \eta$ counterparts. The 't Hooft anomaly does not depend on the choice of the representative in the equivalence class of the defects.

Appendix B discusses these projective algebras and the associated anomaly from the continuum QFT perspective. This anomaly is analogue to the type III anomaly in 1+1d [78, 79, 28]. For convenience, we will also refer to this 2+1d anomaly as a type III anomaly. (See also [80, 81] for similar anomalies in other 2+1d theories.) The existence of such an anomaly forbids the theory from being in a symmetric trivially gapped phase.

## 3.4   Higgs=SPT

We now discuss implication of the type III anomaly and its relation to the Higgs=SPT proposal in [49]. See also [82–85] for related discussions.

We begin with a simplified model by setting $J = \tilde{h} = 0$ in (1.3). The Hamiltonian is [49]

$$H_{\frac{\mathrm{Ising}\times\mathrm{Ising}}{\mathbb{Z}_2}} = -h\sum_v X_v - J'\sum_{\langle v,v'\rangle} Z_v \sigma^z_{\langle v,v'\rangle} Z_{v'} - h'\sum_v X_v \prod_{\ell \ni v} \sigma^x_\ell - \tilde{J}\sum_\ell \sigma^z_\ell - g\sum_f \prod_{\ell \in f} \sigma^z_\ell\,. \tag{3.21}$$

In Appendix C.3, we show that (3.21) is equivalent to two copies of Ising models with the diagonal $\mathbb{Z}_2$ symmetry gauged. Therefore the phase diagram only depends on the two ratios $\tilde{J}/h'$ and $J'/h$ and is shown in Figure 5. By restricting to $h = h', J' = \tilde{J}$, it is a special case of the Hamiltonian (3.1). Therefore on this locus of the parameter space, there is a type III anomaly, and the model cannot be trivially gapped.

Alternatively, the model (3.21) can be viewed as a 2+1d lattice $\mathbb{Z}_2$ gauge theory coupled to Ising matter, with the (electric) Gauss law, i.e., the $h'$ term, enforced energetically. The $Z_v$'s are Ising matter fields with $\mathbb{Z}_2$ gauge charge, and the $\sigma^z_\ell$'s are the $\mathbb{Z}_2$ gauge fields.

The phase where $J'/h \ll 1, \tilde{J}/h' \ll 1$ is a deconfined phase with topological order, with the continuum limit given by the topological $\mathbb{Z}_2$ gauge theory. The magnetic $\mathbb{Z}_2^{(1)}$ 1-form global symmetry is spontaneously broken, but $\mathbb{Z}_2^{(0)}$ is unbroken.

The phase where $J'/h \gg 1, \tilde{J}/h' \gg 1$ is a ferromagnet phase where $\sigma^z_\ell = 1$ and $Z_v Z_{v'} = 1$

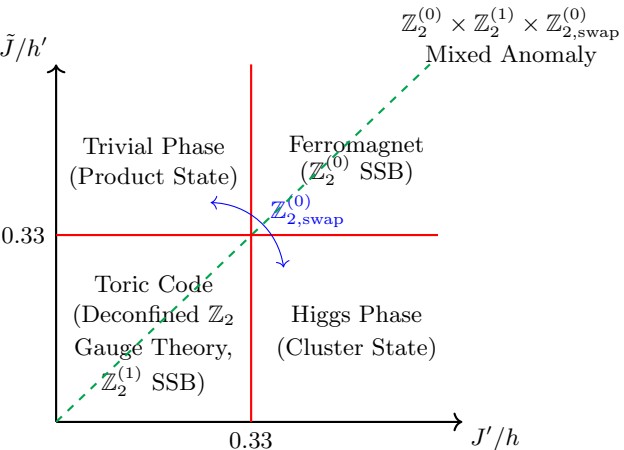

Figure 5: Phase diagram of (3.1). The Hamiltonian on the green dashed line has an anomaly, separating the trivial and Higgs phases.

for neighboring $v, v'$. The $\mathbb{Z}_2^{(0)}$ symmetry is spontaneously broken but the magnetic $\mathbb{Z}_2^{(1)}$ symmetry is unbroken.

In the region where $J'/h \ll 1$ and $\tilde{J}/h' \gg 1$, the $\sigma_\ell^z$ term drives the system to a trivial phase whose ground state is a product state. On the other hand, the phase where $J'/h \gg 1$ and $\tilde{J}/h' \ll 1$ is a Higgs phase, whose ground state is a cluster state that will be discussed in Section 4.1.[25] Both phases are trivially gapped and preserve the $\mathbb{Z}_2^{(0)} \times \mathbb{Z}_2^{(1)}$ symmetry. In both phases, the Wilson line $\prod_{\ell \in \gamma} \sigma_\ell^z \sim 1$, exhibiting perimeter law.

Even though the trivial and the Higgs phases preserve the same generalized symmetry, they are separated by the diagonal line $J'/h = \tilde{J}/h'$ with a type III anomaly where the model cannot be trivially gapped.[26] One has to either cross the gapless point (at $\frac{\tilde{J}}{h'} = \frac{J'}{h} \sim 0.33$), or travel through an intermediate regime of the deconfined or ferromagnet phases.

Finally, we compare our interpretation with the "Higgs=SPT" proposal of [49]. The theories on the two sides of the diagonal line, and in particular, the trivial and the Higgs phases, are related by the $\mathbb{Z}_{2,\mathrm{swap}}^{(0)}$ unitary transformation $V$. It is known that the unitary operator $V$ is an SPT entangler that creates a nontrivial 2+1d $\mathbb{Z}_2^{(0)} \times \mathbb{Z}_2^{(1)}$ SPT state of [58,49] from a product state, and vice versa (see Appendix F). This means that even though these two phases are both trivially gapped on their own, they represent distinct SPT phases [49]. Thus, they cannot be smoothly connected while preserving the $\mathbb{Z}_2^{(0)} \times \mathbb{Z}_2^{(1)}$ symmetry. This is

---

[25]Indeed, in this regime, $Z_v \sigma_{(v,v')}^z Z_{v'} \sim 1$, which is the lattice counterpart of the fact that the gauge field is a pure gauge $A_\mu \sim \partial_\mu \phi$ in a Higgs phase. Here $\phi$ is the Stueckelberg field.

[26]Strictly speaking, only the Hamiltonian with $h = h'$, $J' = \tilde{J}$ has the type III anomaly, but not necessarily those more general Hamiltonians on the diagonal line $J'/h = \tilde{J}/h'$. However, since the phase only depends on these two ratios, the moment we know the special model with $h = h'$, $J' = \tilde{J}$ is nontrivial due to the anomaly, all the other models along the diagonal $J'/h = \tilde{J}/h'$ must be nontrivial as well.

consistent with our explanation above in terms of the mixed anomaly. Indeed, as discussed in Appendix F, one defining feature of the type III anomaly is that it trivializes the relative difference between these two SPT phases. This means that the theory on the diagonal line $J'/h = \tilde{J}/h'$ must have this anomaly.

# 4   Non-invertible SPT phases in 2+1d

In this section we generalize the construction of 1+1d non-invertible SPT phases in [28] to 2+1d. We identify two distinct SPT states protected by the non-invertible swap symmetry $\mathsf{D}$ (which forms 2-Rep$((\mathbb{Z}_2^{(1)} \times \mathbb{Z}_2^{(1)}) \rtimes \mathbb{Z}_2^{(0)})$) in Section 2.

## 4.1   2+1d cluster state as a non-invertible SPT phase

Our starting point is a 2+1d generalization of the 1+1d cluster Hamiltonian [58, 49]:

$$
H_{\text{cluster}} = -\sum_{\langle v,v'\rangle} Z_v \sigma^z_{\langle v,v'\rangle} Z_{v'} - \sum_v X_v \prod_{\ell \ni v} \sigma^x_\ell
$$

$$
= -\sum_{\langle v,v'\rangle} Z\!-\!\sigma^z\!-\!Z - \sum_v \begin{matrix} \sigma^x \\ \sigma^x\!-\!X\!-\!\sigma^x \\ \sigma^x \end{matrix} \,. \tag{4.1}
$$

It is invariant under the $\mathbb{Z}_2^{(0)} \times \mathbb{Z}_2^{(1)}$ symmetry generated by $U = \prod_v X_v$ and $\eta(\gamma) = \prod_{\ell \in \gamma} \sigma^z_\ell$. (We could have also added the magnetic Gauss law term $-g\sum_f \prod_{\ell \in f} \sigma^z_\ell$, but since the latter is a product of the terms in the first sum, it is redundant as far as the ground state is concerned.) We will henceforth refer to (4.1) as the 2+1d cluster Hamiltonian, which is a special case of (2.5).[27] It is also a special case of the Hamiltonian (3.21) deep in the Higgs phase where $J'/h \gg 1, \tilde{J}/h' \ll 1$.

There is a unique ground state, $|\text{cluster}\rangle$, which satisfies

$$
Z_v \sigma^z_{\langle v,v'\rangle} Z_{v'} |\text{cluster}\rangle = |\text{cluster}\rangle \,, \quad X_v \prod_{\ell \ni v} \sigma^x_\ell |\text{cluster}\rangle = |\text{cluster}\rangle \,. \tag{4.2}
$$

From (2.7), one can easily see that this cluster Hamiltonian is furthermore invariant under the non-invertible swap symmetry $\mathsf{D}$. Therefore, $|\text{cluster}\rangle$ is not only a topological phase

---

[27]Since the cluster model is a commuting Pauli Hamiltonian, the phase does not depend on the magnitude of the coupling constants. We have therefore set $J' = h' = 1$ compared to (2.5).

protected by the $\mathbb{Z}_2^{(0)} \times \mathbb{Z}_2^{(1)}$ symmetry, but also by the non-invertible symmetry $\mathsf{D}$. In other words, |cluster⟩ is a 2+1d non-invertible SPT state.

The 't Hooft anomaly of a non-invertible global symmetry is sometimes defined as the incompatibility with a trivially gapped phase [86–88].[28] (See also [66, 92–97].) Since our non-invertible symmetry $\mathsf{D}$ is realized in the cluster model with a non-degenerate gapped ground state, it is non-anomalous in this sense.

A natural question is whether there are other non-invertible SPT states. What are the commuting Pauli Hamiltonians associated with these new SPT states? Note that the product state, which can be taken as the trivial $\mathbb{Z}_2^{(0)} \times \mathbb{Z}_2^{(1)}$ SPT state, is not invariant under $\mathsf{D}$, and is therefore not a non-invertible SPT state.

A similar question has been discussed recently in 1+1d [28], where three SPT phases (including the 1+1d cluster state) for the $\mathrm{Rep}(D_8)$ non-invertible symmetry were identified, matching the classification in the continuum. The key to identifying these new SPT states is the *Kennedy-Tasaki* (KT) transformation [98, 99]. In this section, we first generalize the KT transformation to 2+1d, and then use it to find a new non-invertible SPT phase in 2+1d.

## 4.2   Generalized Kennedy-Tasaki transformation

The KT transformation in 1+1d is defined to be a transformation that maps a $\mathbb{Z}_2^{(0)} \times \mathbb{Z}_2^{(0)}$ symmetry-breaking state to an SPT state, and maps the symmetry-preserving product state to itself. It is then natural to define the *generalized Kennedy-Tasaki* transformation in 2+1d similarly by replacing the symmetry with $\mathbb{Z}_2^{(0)} \times \mathbb{Z}_2^{(1)}$. More specifically, the generalized KT transformation implements the following map between the operators:

$$ X_v \leftrightsquigarrow X_v \,, \qquad Z_v Z_{v'} \leftrightsquigarrow Z_v \sigma^z_{\langle v,v' \rangle} Z_{v'} \,, \qquad \prod_{\ell \ni v} \sigma^x_\ell \leftrightsquigarrow X_v \prod_{\ell \ni v} \sigma^x_\ell \,, \qquad \sigma^z_\ell \leftrightsquigarrow \sigma^z_\ell \,. \quad (4.3) $$

In the continuum, the generalized KT transformation implements a $TST$ transformation, where the $T$ transformation corresponds to stacking the nontrivial $\mathbb{Z}_2^{(0)} \times \mathbb{Z}_2^{(1)}$ SPT phase,[29] and the $S$ transformation corresponds to gauging $\mathbb{Z}_2^{(0)} \times \mathbb{Z}_2^{(1)}$. (Such transformations acting on QFTs are originally discussed in [100] for an ordinary $U(1)$ symmetry. See also [101, 54].)

The generalized KT transformation (4.3) must be implemented by a non-invertible operator. This follows from a proof similar to the one below (2.6). It is straightforward to check

---

[28]See [89–91] for the relation between this definition and the obstruction to gauging the non-invertible symmetry.

[29]In other words, the $T$ transformation adds a counterterm $(-1)^{\int A \cup B}$ to the theory, where $A$ and $B$ are $\mathbb{Z}_2^{(0)}$ and $\mathbb{Z}_2^{(1)}$ background gauge fields, respectively.

that the map (4.3) can be implemented by the following non-invertible operator:

$$\mathsf{KT} = V\mathsf{D}V \,, \tag{4.4}$$

where $V$ is the unitary operator in (3.2), and $\mathsf{D}$ is the non-invertible swap operator in (2.24). For instance,

$$(\mathsf{KT})\, Z_v Z_{v'} = Z_v \sigma^z_{\langle v,v' \rangle} Z_{v'} \,(\mathsf{KT}) \,, \qquad (\mathsf{KT})\, \sigma^x_\ell = X_v \left( \prod_{\ell \ni v} \sigma^x_\ell \right) (\mathsf{KT}) \,. \tag{4.5}$$

We note that the transformation (4.3) exchanges the two Hamiltonians (2.5) and (3.1) we discussed in previous sections.[30] In other words, the non-invertible symmetry $\mathsf{D}$ of (2.5) arises from a mixed anomaly in (3.1) under the twisted gauging of $\mathbb{Z}_2^{(0)} \times \mathbb{Z}_2^{(1)}$. This is an example of a more general phenomenon: non-invertible symmetries can sometimes arise from gauging a non-anomalous subgroup of an anomalous, invertible symmetry [102]. See [103, 104, 81, 105, 106, 15] for examples.

## 4.3 Two non-invertible SPT states

Here we construct a new commuting Pauli Hamiltonian with a non-degenerate ground state invariant under $\mathbb{Z}_2^{(0)} \times \mathbb{Z}_2^{(1)}$ as well as the non-invertible symmetry $\mathsf{D}$. We refer to this Hamiltonian as the cluster$'$ Hamiltonian.

As explained above, the generalized KT transformation exchanges a symmetry-breaking state with an SPT state. Moreover, under this transformation, the non-invertible swap symmetry $\mathsf{D}$ is mapped to an invertible symmetry $\mathbb{Z}_{2,\mathrm{swap}}^{(0)}$ generated by $V$ and vice versa. That is,

$$(\mathsf{KT})\, \mathsf{D} = 2V\,(\mathsf{KT}) \,, \qquad (\mathsf{KT})\, 2V = \mathsf{D}\,(\mathsf{KT}) \tag{4.6}$$

Therefore, different symmetry breaking patterns of $\mathbb{Z}_2^{(0)} \times \mathbb{Z}_2^{(1)} \times \mathbb{Z}_{2,\mathrm{swap}}^{(0)}$ where $\mathbb{Z}_2^{(0)} \times \mathbb{Z}_2^{(1)}$ is spontaneously broken give us different non-invertible SPT states under the generalized KT transformation.

---

[30]More precisely, we need to rename the coefficients $(h, \tilde{h}, J, \tilde{J})$ in (3.1) to be $(h, h', J', J)$ when applying the generalized KT transformation.

Under the generalized KT transformation, the original cluster Hamiltonian becomes[31]

$$\hat{H}_{\text{cluster}} = -\sum_{\langle v,v'\rangle} Z_v Z_{v'} - \sum_v \prod_{\ell\ni v} \sigma_\ell^x - g\sum_f \prod_{\ell\in f} \sigma_\ell^z \,. \tag{4.7}$$

Each ground state is described by a tensor product of a $\mathbb{Z}_2^{(0)}$ symmetry-breaking state on the sites, with a toric code ground state $|\xi\rangle$ in (2.31) on the links. The toric code ground states spontaneously break the $\mathbb{Z}_2^{(1)}$ symmetry. On the other hand, every ground state preserves the $\mathbb{Z}_{2,\text{swap}}^{(0)}$ symmetry.

To find an SPT Hamiltonian different from (4.1), we require its KT image to spontaneously break $\mathbb{Z}_2^{(0)}\times\mathbb{Z}_2^{(1)}$, but preserve a different $\mathbb{Z}_2$ 0-form symmetry. An obvious candidate is the $\mathbb{Z}_2$ symmetry generated by $VU$. The $\mathbb{Z}_2^{(0)}\times\mathbb{Z}_2^{(1)}$ order parameters that commute with $VU$ are

$$\begin{aligned} \mathbb{Z}_2^{(1)}: \quad &\prod_{\ell\in\widehat{\gamma}} \sigma_\ell^x \,, \\ \mathbb{Z}_2^{(0)}: \quad &Y_v \left(1 - \prod_{\ell\ni v} \sigma_\ell^x\right) \,. \end{aligned} \tag{4.8}$$

From these order parameters, one can infer that the Hamiltonian whose ground states spontaneously break $\mathbb{Z}_2^{(0)}\times\mathbb{Z}_2^{(1)}$ but preserve the $\mathbb{Z}_2$ generated by $VU$ is

$$\hat{H}_{\text{cluster}'} = \sum_v \prod_{\ell\ni v} \sigma_\ell^x - \sum_{\langle v,v'\rangle} Y_v Y_{v'} \left(1 + \prod_{\ell\ni v} \sigma_\ell^x \prod_{\ell'\ni v'} \sigma_{\ell'}^x\right) - g\sum_f \prod_{\ell\in f} \sigma_\ell^z \,. \tag{4.9}$$

This is a commuting Pauli Hamiltonian, and its ground states satisfy

$$Y_v Y_{v'} = 1\,, \quad \prod_{\ell\ni v} \sigma_\ell^x = -1\,, \quad \prod_{\ell\in f} \sigma_\ell^z = 1\,, \tag{4.10}$$

where $v, v'$ are neighboring vertices. The condition $\prod_{\ell\ni v}\sigma_\ell^x = -1$ can only be satisfied if $L_x L_y$ is even.[32] We assume $L_x$ to be even below, while the even $L_y$ case is similar. Each ground state is described by a tensor product of a $\mathbb{Z}_2^{(0)}$ symmetry-breaking state on the sites satisfying $Y_v = 1$ or $Y_v = -1$ for all $v$, with a ground state of the *odd toric code* Hamiltonian.

---

[31]The generalized KT transformation is only unambiguous in the constrained Hilbert space $\widetilde{\mathcal{H}}$ where the magnetic Gauss law is imposed $\prod_{\ell\in f}\sigma_\ell^z = 1$. We have added back the term $-g\sum_f\prod_{\ell\in f}\sigma_\ell^z$ in $\hat{H}_{\text{cluster}}$ to enforce the magnetic Gauss law energetically.

[32]To see this, consider the product over all $v$ on both sides of $\prod_{\ell\ni v}\sigma_\ell^x = -1$. The left hand side is 1 while the right hand side is $(-1)^{L_x L_y}$. Hence $L_x L_y$ is even.

The latter is

$$H_{\text{odd TC}} = KH_{\text{TC}}K = \sum_v \prod_{\ell \ni v} \sigma_\ell^x - g \sum_f \prod_{\ell \in f} \sigma_\ell^z, \tag{4.11}$$

where

$$K = \prod_{\substack{i=2 \\ i \text{ even}}}^{L_x} \prod_{j=1}^{L_y} \sigma_{i+\frac{1}{2},j}^z. \tag{4.12}$$

The odd toric code describes the deconfined phase of the *odd $\mathbb{Z}_2$ gauge theory* [107]. It also has four ground states $K|\xi\rangle$, where $\xi \in \{0,1\}^2$, and each of them spontaneously breaks the $\mathbb{Z}_2^{(1)}$ symmetry. The ground states of the total system on the sites and links preserve $VU$.

The Hamiltonian for the other non-invertible SPT phase is then obtained by performing the generalized KT transformation on (4.9),

$$
\begin{aligned}
H_{\text{cluster}'} &= \sum_v X_v \prod_{\ell \ni v} \sigma_\ell^x - \sum_{\langle v,v' \rangle} Y_v \sigma_{\langle v,v' \rangle}^z Y_{v'} - \sum_{\langle v,v' \rangle} Z_v \left( \prod_{\ell \ni v} \sigma_\ell^x \right) \sigma_{\langle v,v' \rangle}^z \left( \prod_{\ell' \ni v'} \sigma_{\ell'}^x \right) Z_{v'} \\
&= \sum_v \begin{matrix} \sigma^{x} \\ \sigma^x\!-\!X\!-\!\sigma^x \\ \sigma^{x} \end{matrix} - \sum_{\langle v,v' \rangle} Y\!-\!\sigma^z\!-\!Y + \sum_{\langle v,v' \rangle} \begin{matrix} \sigma^{x} \quad \sigma^{x} \\ \sigma^x\!-\!Z\!-\!\sigma^z\!-\!Z\!-\!\sigma^x \\ \sigma^{x} \quad \sigma^{x} \end{matrix} \,.
\end{aligned}
\tag{4.13}
$$

Note that the third term is the product of the terms from the first two sums, and is needed for the Hamiltonian $H_{\text{cluster}'}$ to commute with $\mathsf{D}$. This SPT Hamiltonian $H_{\text{cluster}'}$ has a unique ground state $|\text{cluster}'\rangle$ satisfying

$$X_v \prod_{\ell \ni v} \sigma_\ell^x |\text{cluster}'\rangle = -|\text{cluster}'\rangle, \quad Y_v \sigma_{\langle v,v' \rangle}^z Y_{v'} |\text{cluster}'\rangle = |\text{cluster}'\rangle. \tag{4.14}$$

We leave the complete classification of the SPT phases for this noninvertible symmetry, as well as the interfaces between them, for future investigations.

# 5 Conclusion and outlook

In this work, we studied the exact lattice generalized symmetries and anomalies of the lattice $\mathbb{Z}_2$ gauge theory coupled to Ising matter (1.3).

We found that in different limits of the parameter space $\{h, \tilde{h}, h', J, \tilde{J}, J', g\}$, there are

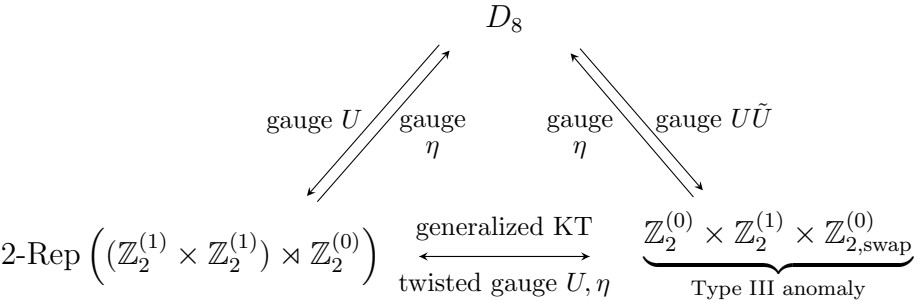

Figure 6: The generalized symmetries in the three models are related by (twisted) gauging.

exact, non-invertible/higher-form symmetries and anomalies of various kinds, summarized in Table 1. We also found the exactly solvable lattice models of two distinct SPT phases of a non-invertible symmetry.

| Hamiltonian | Symmetry | Operators | Anomaly | Parameters |
|---|---|---|---|---|
| $H_{\mathrm{AT}}$ (2.9) | $D_8$ | $U, \tilde{U}, S$ (2.11) | No | N.A. |
| $H_{\mathrm{noninv}}$ (2.5) | 2-Rep$((\mathbb{Z}_2^{(1)} \times \mathbb{Z}_2^{(1)}) \rtimes \mathbb{Z}_2^{(0)})$ | $U, \eta(\gamma), \mathsf{D}$ (2.36) | No | $h = \tilde{h}, J = \tilde{J}$ |
| $H_{\mathrm{anom}}$ (3.1) | $\mathbb{Z}_2^{(0)} \times \mathbb{Z}_2^{(1)} \times \mathbb{Z}_{2,\mathrm{swap}}^{(0)}$ | $U, \eta(\gamma), V$ | Type III | $h = h', J' = \tilde{J}$ |

Table 1: Summary of the models. Here each of $U, \tilde{U}, S, V$ generates an ordinary $\mathbb{Z}_2^{(0)}$ symmetry, $\eta(\gamma)$ generates $\mathbb{Z}_2^{(1)}$ 1-form symmetry, and $\mathsf{D}$ is non-invertible.

These models are not unrelated; rather, they are connected by (twisted) gauging. In Section 2.2, we showed that $H_{\mathrm{noninv}}$ is obtained from the Ashkin-Teller-like Hamiltonian $H_{\mathrm{AT}}$ (with $(h, J) = (\tilde{h}, \tilde{J})$) by gauging an ordinary $\mathbb{Z}_2^{(0)}$ symmetry generated by $\tilde{U}$. It is also straightforward to check that $H_{\mathrm{anom}}$ is obtained from $H_{\mathrm{AT}}$ by gauging a different $\mathbb{Z}_2^{(0)}$ symmetry generated by $U\tilde{U}$.[33] Moreover, in Section 4.2, we found that $H_{\mathrm{noninv}}$ and $H_{\mathrm{anom}}$ are related (up to renaming the parameters) by the generalized KT transformation — a twisted gauging of $\mathbb{Z}_2^{(0)} \times \mathbb{Z}_2^{(1)}$. Their schematic relations are summarized in Figure 6. Our lattice models realize three of the continuum QFTs in the duality web of [60, 61].

We can consider an even more special limit in the parameter space, which is the intersection of those for $H_{\mathrm{noninv}}$ and $H_{\mathrm{anom}}$. Namely

$$h = \tilde{h} = h', \quad J = \tilde{J} = J'. \tag{5.1}$$

---

[33]We explain in Appendix C.3 how to gauge $U\tilde{U}$. To identify the gauged $H_{\mathrm{AT}}$ with $H_{\mathrm{anom}}$, one needs to replace the parameters $(h, J, \tilde{h}, \tilde{J}, h', J')$ in the former by $(h, \tilde{J}, h, \tilde{J}, \tilde{h}, J)$.

The corresponding Hamiltonian simultaneously enjoys the non-invertible symmetry $\mathsf{D}$ and the anomalous invertible symmetry $\mathbb{Z}_2^{(0)} \times \mathbb{Z}_2^{(1)} \times \mathbb{Z}_{2,\text{swap}}^{(0)}$ generated by $V, U, \eta$. They satisfy

$$\mathsf{D}V\mathsf{D} = 2V\mathsf{D}V \, . \tag{5.2}$$

The analogous relation in 1+1d was discussed in [28].

The model has many non-invertible symmetries in addition to $\mathsf{D}$. One of them, $\mathsf{KT} = V\mathsf{D}V$ is non-anomalous in the sense that it is compatible with the trivial paramagnet Hamiltonian $-\sum_v X_v - \sum_\ell \sigma_\ell^z$. Another non-invertible operator $\mathsf{D}V$ is a triality operator, which is analogous to the triality operators in continuum 1+1d and $3+1$d field theories discussed in [108, 66, 109]. It would be interesting to explore its anomaly and constraints on the phase diagram.

Finally, we list some interesting future directions.

1. Our models are arguably the simplest family of lattice models hosting exact 1-form and non-invertible symmetries. Lattice models with a general finite fusion 2-category symmetries have been constructed, known as the fusion surface models [19]. Generically, these models are defined on non-tensor product Hilbert spaces. In contrast, our lattice model is defined on a tensor product Hilbert space and realizes the fusion 2-category 2-Rep$((\mathbb{Z}_2^{(1)} \times \mathbb{Z}_2^{(1)}) \rtimes \mathbb{Z}_2^{(0)})$. It would be interesting to understand how our models are related to the fusion surface models.

2. It would be interesting to study the lattice defects associated with the non-invertible operator $\mathsf{D}$ and the condensation operator $\mathsf{C}$. See [9, 10, 12, 13, 16] for the defect of the non-invertible Kramers-Wannier symmetry in 1+1d.

3. Given that the non-invertible symmetry generated by $U, \eta, \mathsf{D}$ is non-anomalous, one should be able to gauge it. How to gauge a non-invertible symmetry on the lattice?

4. Mathematically, the non-invertible SPT phases of a fusion 2-category are classified by the fiber 2-functors. It would be interesting to compare our 2+1d non-invertible SPT states $|\text{cluster}\rangle$ and $|\text{cluster}'\rangle$ with the mathematical classification in [110].

5. What can live at the 1+1d interface between the two 2+1d non-invertible SPT states?

6. Generalization to higher dimensions. See [111, 112, 88, 68] for non-invertible defects and operators in $3+1$d.

# Acknowledgements

We thank Ashvin Vishwanath for discussions which initiated this project. We are grateful to Weiguang Cao, Pranay Gorantla, Linhao Li, Sahand Seifnashri, Nathanan Tantivasadakarn, Ruben Verresen, and Matthew Yu for interesting discussions. SHS thanks Pranay Gorantla and Nathanan Tantivasadakarn for collaboration on a related project [68] about the non-invertible Wegner duality symmetry in 3+1d lattice $\mathbb{Z}_2$ gauge theory. We thank Luisa Eck for comments on the manuscript. The work of SHS was supported in part by NSF grant PHY-2210182. The authors of this paper were ordered alphabetically.

# A  Non-invertible symmetries in 2+1d QFT

Following [88, 66, 81], we review the continuum analog of the non-invertible symmetry operator found in Section 2.

## A.1  Half gauging

Let $\mathcal{T}$ be a QFT, in arbitrary spacetime dimensions, which has a finite (internal) symmetry $G$ that can be gauged. One can form a topological interface between $\mathcal{T}$ and the gauged theory $\mathcal{T}/G$ by gauging $G$ in only half of spacetime, say $x > 0$, and then imposing the topological Dirichlet boundary condition for the gauge field (or a generalization thereof) at $x = 0$. We call such a topological interface a *half-gauging interface*. Here, $G$ can be an ordinary finite group symmetry [108], a higher-form symmetry [88, 66], or a non-invertible symmetry [91, 113, 114].[34] In this Appendix, we focus on the case where $G$ is invertible, but it can still include higher-form symmetries.

Suppose that the theory $\mathcal{T}$ obeys the special property that it is isomorphic to its gauged theory $\mathcal{T}/G$:

$$\mathcal{T} \cong \mathcal{T}/G \,. \tag{A.1}$$

In particular, the original global symmetry $G$ is identified with the dual global symmetry $\widehat{G}$, i.e., $G \cong \widehat{G}$ [115, 54, 102].[35] When $\mathcal{T}$ is self-dual under gauging in the above sense, the half-gauging interface can be composed with an invertible topological interface between $\mathcal{T}$ and $\mathcal{T}/G$ implementing the isomorphism (A.1). The result is a topological defect in a single

---

[34] When $G$ is invertible, $G$ can be gauged if and only if it is free on an 't Hooft anomaly, by definition. When $G$ is non-invertible, both the anomaly and gauging need to be carefully defined. See [90, Section 3] for discussions on the relation between anomalies and gauging for non-invertible symmetries.

[35] For instance, the dual global symmetry of a $q$-form, finite group symmetry $G^{(q)}$ is $\widehat{G}^{(d-q-2)}$ in $d$ spacetime dimensions, where $\widehat{G} = \text{Hom}(G, U(1))$ is the Pontryagin dual of $G$.

theory $\mathcal{T}$, known as a duality defect or a duality operator.[36] Such a duality defect generally does not have an inverse and defines a non-invertible symmetry of the theory $\mathcal{T}$.

The simplest example of a duality defect is the Kramers-Wannier defect line in the 1+1d Ising CFT [116, 117, 86, 118], where the symmetry $G$ is the ordinary $\mathbb{Z}_2^{(0)}$ spin-flip symmetry, and the isomorphism (A.1) is given by the Kramers-Wannier duality.

## A.2   Gauging $\mathbb{Z}_2^{(0)} \times \mathbb{Z}_2^{(1)}$ in 2+1d

The continuum counterpart of our lattice discussions in Section 2 corresponds to choosing a 2+1d QFT $\mathcal{T}$ with a $G = \mathbb{Z}_2^{(0)} \times \mathbb{Z}_2^{(1)}$ symmetry, such that it is invariant under gauging,

$$\mathcal{T}/(\mathbb{Z}_2^{(0)} \times \mathbb{Z}_2^{(1)}) \cong \mathcal{T}. \tag{A.2}$$

The gauged theory has a dual global symmetry $\widehat{\mathbb{Z}}_2^{(1)} \times \widehat{\mathbb{Z}}_2^{(0)}$ [115, 54, 102], which is identified with $G$ under the isomorphism.[37] We denote the 0-form symmetry operator as $\mathcal{U}$, and the 1-form symmetry operator as $\eta$. Here $\mathcal{U}$ and $\eta$ are topological surface and line operators in spacetime, respectively.

In particular, as commented at the end of Section 2.1, a special class of such QFTs corresponds to taking $\mathcal{T} = \mathcal{Q} \times \frac{\mathcal{Q}}{\mathbb{Z}_2^{(0)}}$ where $\mathcal{Q}$ is an arbitrary QFT with a non-anomalous $\mathbb{Z}_2^{(0)}$ 0-form symmetry, and $\mathcal{Q}/\mathbb{Z}_2^{(0)}$ consequently has a dual 1-form $\mathbb{Z}_2^{(1)}$ symmetry. Any QFT $\mathcal{T}$ of this form is self-dual under gauging $\mathbb{Z}_2^{(0)} \times \mathbb{Z}_2^{(1)}$, which exchanges $\mathcal{Q}$ with $\mathcal{Q}/\mathbb{Z}_2^{(0)}$. In Section 2, the model (2.5) with $h' = J' = 0$ is a lattice counterpart of $\mathcal{T}$.

## A.3   Duality and condensation defects

The continuum theory $\mathcal{T}$ has a non-invertible duality defect, which we denote as $\mathcal{D}$, coming from gauging the $\mathbb{Z}_2^{(0)} \times \mathbb{Z}_2^{(1)}$ symmetry in half of spacetime. The duality defect obeys a

---

[36]Since the continuum discussion is mostly in Euclidean signature, we use the terms "operator" and "defect" interchangeably.

[37]In 2+1d, the dual symmetry of a $q$-form symmetry is a $(1-q)$-form symmetry. Hence a QFT $\mathcal{T}$ cannot be self-dual if $G$ is a $q$-form symmetry of a fixed form degree. The simplest nontrivial self-dual example in 2+1d involves taking $G$ to be a product group of both 0- and 1-form symmetries.

fusion algebra [88, 66]

$$\mathcal{D}^2 = (1 + \mathcal{U}) \left( \frac{1}{|H^0(\Sigma, \mathbb{Z}_2)|} \sum_{\gamma \in H_1(\Sigma, \mathbb{Z}_2)} \eta(\gamma) \right), \tag{A.3}$$

$$\mathcal{D}\mathcal{U} = \mathcal{U}\mathcal{D} = \mathcal{D}, \quad \mathcal{D}\eta = \eta\mathcal{D} = \mathcal{D}.$$

Here, $\Sigma$ is the support of the duality defect $\mathcal{D}$ which can be an arbitrary 2-dimensional closed submanifold in spacetime. On the right-hand side of the first equation, we have the continuum condensation defects defined by [119, 65, 66]

$$\mathcal{C} \equiv (1 + \mathcal{U})\mathcal{C}_\eta,$$

$$\mathcal{C}_\eta \equiv \frac{1}{|H^0(\Sigma, \mathbb{Z}_2)|} \sum_{\gamma \in H_1(\Sigma, \mathbb{Z}_2)} \eta(\gamma). \tag{A.4}$$

The condensation defect $\mathcal{C}$ for the $\mathbb{Z}_2^{(0)} \times \mathbb{Z}_2^{(1)}$ symmetries can absorb the invertible symmetry defects in the following sense:

$$\mathcal{C}\mathcal{U} = \mathcal{U}\mathcal{C} = \mathcal{C}, \quad \mathcal{C}\eta = \eta\mathcal{C} = \mathcal{C}. \tag{A.5}$$

The two (codimension-1) non-invertible defects $\mathcal{D}$ and $\mathcal{C}$ obey [65, 66]

$$\mathcal{D}^2 = \mathcal{C}, \quad \mathcal{C}^2 = 2(\mathcal{Z}_2)\mathcal{C}, \quad \mathcal{D}\mathcal{C} = \mathcal{C}\mathcal{D} = 2(\mathcal{Z}_2)\mathcal{D}, \tag{A.6}$$

where $\mathcal{Z}_2$ denotes the partition function of 1+1d $\mathbb{Z}_2$ gauge theory on the surface $\Sigma$. When $\Sigma$ is a torus, we have $\mathcal{Z}_2 = 2$, matching the lattice algebra in (2.37).

## A.4    2-Rep$((\mathbb{Z}_2^{(1)} \times \mathbb{Z}_2^{(1)}) \rtimes \mathbb{Z}_2^{(0)})$ from gauging a subgroup of $D_8$

An alternative way to understand the duality defect $\mathcal{D}$ in $\mathcal{T}$ is as follows. We start with another theory $\mathcal{T}'$, which is assumed to have an ordinary (0-form, invertible) non-anomalous $D_8$ symmetry. The model (2.9) (with $(h, J) = (\tilde{h}, \tilde{J})$) is the lattice counterpart of $\mathcal{T}'$. The group $D_8$ can be decomposed as

$$0 \to \mathbb{Z}_2^{(0)} \times \tilde{\mathbb{Z}}_2^{(0)} \to D_8 \to \mathbb{Z}_{2,\text{swap}}^{(0)} \to 0. \tag{A.7}$$

This is a split group extension, where $\mathbb{Z}_{2,\text{swap}}^{(0)}$ acts by swapping the two factors in $\mathbb{Z}_2^{(0)} \times \tilde{\mathbb{Z}}_2^{(0)}$. The generators of $\mathbb{Z}_2^{(0)}, \tilde{\mathbb{Z}}_2^{(0)}$ and $\mathbb{Z}_{2,\text{swap}}^{(0)}$ are denoted as $\mathcal{U}, \tilde{\mathcal{U}}$ and $\mathcal{S}$ respectively. They satisfy

the standard $D_8$ group multiplication law

$$\mathcal{U}^2 = \tilde{\mathcal{U}}^2 = \mathcal{S}^2 = 1, \quad \mathcal{U}\mathcal{S} = \mathcal{S}\tilde{\mathcal{U}}, \quad \mathcal{S}\mathcal{U} = \tilde{\mathcal{U}}\mathcal{S}, \quad \mathcal{U}\tilde{\mathcal{U}} = \tilde{\mathcal{U}}\mathcal{U}. \tag{A.8}$$

Denote the background fields for $\mathbb{Z}_2^{(0)}, \tilde{\mathbb{Z}}_2^{(0)}$ as $A, \tilde{A}$, and the partition function of the theory on a closed 3-manifold $M_3$ in the presence of background gauge fields as $\mathcal{Z}_{\mathcal{T}'}[M_3; A, \tilde{A}]$. Since $\mathcal{T}'$ is $D_8$ invariant, we have

$$\mathcal{Z}_{\mathcal{T}'}[M_3; A, \tilde{A}] = \mathcal{Z}_{\mathcal{T}'}[M_3; \tilde{A}, A]. \tag{A.9}$$

The theory $\mathcal{T}$ can be constructed from $\mathcal{T}'$ via gauging $\tilde{\mathbb{Z}}_2^{(0)}$:

$$\mathcal{T} = \mathcal{T}'/\tilde{\mathbb{Z}}_2^{(0)}. \tag{A.10}$$

The theory $\mathcal{T}$ has $\mathbb{Z}_2^{(0)}$ symmetry that descends from $\mathcal{T}'$, as well as a dual $\mathbb{Z}_2^{(1)}$ 1-form symmetry from gauging.

What happens to $\mathbb{Z}_{2,\text{swap}}^{(0)}$? Below we argue that $\mathbb{Z}_{2,\text{swap}}^{(0)}$ in $\mathcal{T}'$ becomes a non-invertible symmetry in $\mathcal{T}$. This can be easily seen from the partition function of $\mathcal{T}$. We have[38]

$$\mathcal{Z}_{\mathcal{T}}[M_3; A, B] = \sum_a \mathcal{Z}_{\mathcal{T}'}[M_3; A, a](-1)^{\int_{M_3} a \cup B}, \tag{A.11}$$

where $B$ is the 2-form background field for the 1-form symmetry. Then gauging $\mathbb{Z}_2^{(0)} \times \mathbb{Z}_2^{(1)}$ of $\mathcal{T}$ gives

$$\begin{aligned}
\sum_{a,b} \mathcal{Z}_{\mathcal{T}}[M_3; a, b](-1)^{\int_{M_3} a \cup B + b \cup A} &= \sum_{a,b,a'} \mathcal{Z}_{\mathcal{T}'}[M_3; a, a'](-1)^{\int_{M_3} a' \cup b + a \cup B + b \cup A} \\
&= \sum_a \mathcal{Z}_{\mathcal{T}'}[M_3; a, A](-1)^{\int_{M_3} a \cup B} \\
&= \sum_a \mathcal{Z}_{\mathcal{T}'}[M_3; A, a](-1)^{\int_{M_3} a \cup B} \\
&= \mathcal{Z}_{\mathcal{T}}[M_3; A, B].
\end{aligned} \tag{A.12}$$

In the third line, we used the invertible swap symmetry (A.9). This shows that $\mathcal{T}$ is self-dual under gauging $\mathbb{Z}_2^{(0)} \times \mathbb{Z}_2^{(1)}$, which leads to a non-invertible defect by half gauging.

This construction we outlined here is the continuum counterpart of the lattice non-invertible operator in Section 2. This non-invertible symmetry in the continuum forms the fusion 2-category 2-Rep$((\mathbb{Z}_2^{(1)} \times \mathbb{Z}_2^{(1)}) \rtimes \mathbb{Z}_2^{(0)})$ [60, 61].

---

[38]For simplicity, we omit overall normalizations of the partition functions.

# B  Type III anomaly involving a 1-form symmetry in 2+1d QFT

In this appendix we briefly explain the continuum QFT interpretation for the mixed anomaly discussed in Section 3. Our explanation follows that in [80, Appendix B], and more general discussions can be found in [102].

## B.1  Mixed anomaly from gauging a subgroup of $D_8$

Let $\mathcal{T}'$ be an arbitrary 2+1d QFT which has a non-anomalous $D_8$ (0-form) symmetry. The Ashkin-Teller-like model (2.9) (with $(h, \tilde{J}) = (h', J')$) is the lattice counterpart of $\mathcal{T}'$.[39]

The dihedral group $D_8$ is given by a central extension

$$0 \to \mathbb{Z}_{2,\text{diag}}^{(0)} \to D_8 \to \mathbb{Z}_{2,\text{swap}}^{(0)} \times \mathbb{Z}_2^{(0)} \to 0 \,. \tag{B.1}$$

The extension is characterized by the nontrivial element in $H^2(\mathbb{Z}_{2,\text{swap}}^{(0)} \times \mathbb{Z}_2^{(0)}, \mathbb{Z}_{2,\text{diag}}^{(0)}) \cong \mathbb{Z}_2$. This central extension is not to be confused with the split extension (A.7). The generators for $\mathbb{Z}_{2,\text{swap}}^{(0)}$ and $\mathbb{Z}_2^{(0)}$ are $\mathcal{S}$ and $\mathcal{U}$, respectively. The center $\mathbb{Z}_{2,\text{diag}}^{(0)}$ of $D_8$ is the diagonal subgroup of $\mathbb{Z}_2^{(0)} \times \tilde{\mathbb{Z}}_2^{(0)}$, generated by $\mathcal{U}\tilde{\mathcal{U}}$ which commutes with both $\mathcal{S}$ and $\mathcal{U}$. Using (A.8), it is easy to check that $\mathcal{S}$ and $\mathcal{U}$ commute up to $\mathcal{U}\tilde{\mathcal{U}}$,

$$\mathcal{S}\mathcal{U} = (\mathcal{U}\tilde{\mathcal{U}})\mathcal{U}\mathcal{S} \,. \tag{B.2}$$

This means that the central extension is non-trivial.

We can express the background gauge field for the $D_8$ symmetry using three $\mathbb{Z}_2$-valued background gauge fields [80, 78]. We denote the background fields of $\mathbb{Z}_{2,\text{swap}}^{(0)}$, $\mathbb{Z}_2^{(0)}$, $\mathbb{Z}_{2,\text{diag}}^{(0)}$ as $C, A, A'$, respectively. The background fields $C$ and $A$ are both $\mathbb{Z}_2$-valued 1-cocycles, $\delta C = \delta A = 0$, whereas $A'$ is a $\mathbb{Z}_2$-valued 1-cochain satisfying

$$\delta A' = C \cup A \,. \tag{B.3}$$

Equations (B.2) and (B.3) are different, but equivalent, manifestations of the non-trivial central extension (B.1). We denote the partition function of the theory $\mathcal{T}'$ on a closed manifold $M_3$ coupled to these background gauge fields as $\mathcal{Z}_{\mathcal{T}'}[M_3; A, C, A']$.

---

[39]Note that the theory $\mathcal{T}'$ here and the one in Appendix A correspond to different limits in the parameter space of the lattice Hamiltonian (2.9).

Our goal is to understand the symmetries and anomalies of the theory

$$\mathcal{T}'' = \mathcal{T}'/\mathbb{Z}_{2,\mathrm{diag}}^{(0)}. \tag{B.4}$$

To begin with, the 0-form symmetry that survives in the theory $\mathcal{T}''$ after gauging $\mathbb{Z}_{2,\mathrm{diag}}^{(0)}$ is $D_8/\mathbb{Z}_{2,\mathrm{diag}}^{(0)} \cong \mathbb{Z}_{2,\mathrm{swap}}^{(0)} \times \mathbb{Z}_2^{(0)}$, whose corresponding background gauge fields are $C$ and $A$. In addition, there is a dual $\mathbb{Z}_2^{(1)}$ 1-form symmetry coming from gauging $\mathbb{Z}_{2,\mathrm{diag}}^{(0)}$. We denote the background gauge field for the dual $\mathbb{Z}_2^{(1)}$ 1-form symmetry as $B$, which is a $\mathbb{Z}_2$-valued 2-cocycle.

The partition function of the gauged theory $\mathcal{T}''$ on a closed manifold $M_3$ coupled to the various background gauge fields is given by[40]

$$\mathcal{Z}_{\mathcal{T}''}[M_3; A, C, B] = \sum_{a'} \mathcal{Z}_{\mathcal{T}'}[M_3; A, C, a'](-1)^{\int_{M_3} B \cup a'}, \tag{B.5}$$

where the summation is over the $\mathbb{Z}_2$ gauge field configurations of $a'$ satisfying $\delta a' = C \cup A$.

Now, we perform a background gauge transformation $B \to B + \delta\lambda$ where $\lambda$ is a $\mathbb{Z}_2$-valued 1-cochain. Since $\delta a' = C \cup A$, the partition function of $\mathcal{T}''$ is not invariant under such a background gauge transformation. Rather, we have

$$\mathcal{Z}_{\mathcal{T}''}[M_3; A, C, B + \delta\lambda] = \mathcal{Z}_{\mathcal{T}''}[M_3; A, C, B](-1)^{\int_{M_3} \lambda \cup C \cup A}. \tag{B.6}$$

The anomalous phase is nontrivial only if the other two background gauge fields $A$ and $C$ are activated. Therefore, the theory $\mathcal{T}''$ has a mixed 't Hooft anomaly involving all three symmetries $\mathbb{Z}_2^{(0)} \times \mathbb{Z}_{2,\mathrm{swap}}^{(0)} \times \mathbb{Z}_2^{(1)}$, which is analogous to the type III anomaly in 1+1d for a $(\mathbb{Z}_2^{(0)})^3$ symmetry.

The partition function can be made gauge invariant by coupling it to a $3+1$d topological term:

$$\mathcal{Z}_{\mathcal{T}''}[M_3; A, C, B](-1)^{\int_{M_4} B \cup C \cup A}, \tag{B.7}$$

where $\partial M_4 = M_3$. Hence, the inflow action for the anomaly is given by $(-1)^{\int_{M_4} B \cup C \cup A}$.

---

[40]There are two ways to gauge the $\mathbb{Z}_2^{(0)}$ 0-form symmetry in 2+1d, which differ by the choice of discrete torsion valued in $H^3(\mathbb{Z}_2, U(1)) \cong \mathbb{Z}_2$. Here we make a particular choice, but our discussion in this appendix is valid independent of the choice of discrete torsion.

## B.2 Projective algebras from the type III anomaly

We now discuss the consequences of this mixed anomaly, and relate them to the various projective algebras in Section 3. We denote the $\mathbb{Z}_2^{(0)}, \mathbb{Z}_{2,\text{swap}}^{(0)}, \mathbb{Z}_2^{(1)}$ defects in $\mathcal{T}'$ as $\mathcal{U}, \mathcal{V}$ and $\eta$, respectively. $\mathcal{U}$ and $\mathcal{V}$ are topological surface operators, and $\eta$ is a topological line operator. They correspond to the lattice operators $U, V, \eta$ in Section 3.3.

First, we insert the 1-form symmetry defect $\eta$ along a nontrivial 1-cycle $\gamma$ in spacetime $M_3$. Along the worldline $\gamma$, there is an anomaly inflow for the other two $\mathbb{Z}_2^{(0)}$ 0-form symmetries. Specifically, the background gauge field $B$ is given by the Poincaré dual of $\gamma$ (and extended to the bulk $M_4$), and the anomaly inflow action reduces to

$$(-1)^{\int_{M_4} B \cup C \cup A} = (-1)^{\int_{M_2} C \cup A} , \tag{B.8}$$

where $\partial M_2 = \gamma$. The righthand side is the inflow action for an anomaly of a $\mathbb{Z}_2^{(0)} \times \mathbb{Z}_{2,\text{swap}}^{(0)}$ symmetry in quantum mechanics. Such an anomaly in quantum mechanics manifests itself as a projective algebra of the two operators $\mathcal{U}$ and $\mathcal{V}$. We conclude that in $\mathcal{T}''$, the two symmetry operators $\mathcal{U}$ and $\mathcal{V}$ are realized projectively in the presence of a 1-form symmetry defect $\eta$. This projective algebra is precisely the one that we found on the lattice in (3.16). The other lattice projective algebras (3.18) and (3.20) similarly have their continuum counterparts.

## B.3 Modular constraints in 2+1d

In this subsection we explain why the presence of this anomaly implies that the low energy phase cannot be a symmetric, trivially gapped phase, i.e., an invertible TQFT with the given symmetries. Our argument can be view as a 2+1d version of the standard modular covariance argument in 1+1d.

Suppose on the contrary that such an invertible TQFT exists. We place this theory on a Euclidean three-dimensional torus $T^3$, and wrap the 1-form symmetry defect $\eta$ around one of the cycles. If we view this cycle that $\eta$ wraps around as the Euclidean time direction, then the partition function of the invertible TQFT on this geometry computes the dimension of the $T^2$ Hilbert space twisted by the insertion of a single $\eta$ defect. As we discussed, the anomaly implies that the $\mathcal{U}$ and $\mathcal{V}$ operators act projectively on this twisted Hilbert space. Hence, such a partition function, being equal to the dimension of the twisted Hilbert space, is greater than 1.

On the other hand, if we view one of the other two cycles as the Euclidean time circle, then the same partition function corresponds to the expectation value of the 1-form symmetry operator $\eta$ acting on the unique ground state in the untwisted $T^2$ Hilbert space of the theory.

By assumption, this unique ground state is symmetric under $\eta$, and hence the expectation value is 1. This is in contradiction with the previous conclusion that the partition function on this geometry must be greater than 1. This proves that there does not exists a symmetric invertible TQFT that can match the anomaly (B.7), and hence the theory $\mathcal{T}''$ cannot be trivially gapped while preserving the symmetries.

# C   Gauging 0- and 1-form lattice symmetries

Here, we gauge the $\mathbb{Z}_2^{(0)}$ symmetry of the Ising model to obtain $\frac{\text{Ising}}{\mathbb{Z}_2}$ and show that it is equivalent to a $\mathbb{Z}_2$ gauge theory. We will then gauge the 1-form symmetry $\mathbb{Z}_2^{(1)}$ in the $\mathbb{Z}_2$ gauge theory (or equivalently $\frac{\text{Ising}}{\mathbb{Z}_2}$) and show that it results in the Ising model. Throughout we work on a square lattice with periodic boundary conditions, i.e. a torus.

## C.1   Gauging the 0-form symmetry

The Hilbert space of Ising model in 2+1d consists of qubits on the sites with the following Hamiltonian:

$$H_{\text{Ising}} = -J \sum_{\langle v,v' \rangle} Z_v Z_{v'} - h \sum_v X_v \,. \tag{C.1}$$

There is a $\mathbb{Z}_2^{(0)}$ symmetry generated by $U = \prod_v X_v$ which transforms $Z_v \to -Z_v$ and leaves $X_v$ invariant. In order to gauge this symmetry, we introduce $\mathbb{Z}_2$ 1-form gauge fields $\widehat{\sigma}_\ell^z$ and the corresponding "electric" field $\widehat{\sigma}_\ell^x$ on the links, where $\widehat{\sigma}_\ell^{z,x}$ are the Pauli operators on the link $\ell$. We couple these fields to the Hamiltonian as follows

$$H_{\text{gauge}} = -J \sum_{\langle v,v' \rangle} Z_v \widehat{\sigma}_{v,v'}^z Z_{v'} - h \sum_v X_v - g \sum_f \prod_{\ell \in f} \widehat{\sigma}_\ell^z \,, \tag{C.2}$$

where we have added the magnetic flux term $g \sum_f \prod_{\ell \in f} \widehat{\sigma}_\ell^z$. The corresponding Gauss law reads

$$G_v = X_v \prod_{\ell \ni v} \widehat{\sigma}_\ell^x = 1 \,. \tag{C.3}$$

We impose this Gauss law condition strictly. Gauging the $\mathbb{Z}_2^{(0)}$ symmetry leads to a dual 1-form $\mathbb{Z}_2^{(1)}$ symmetry generated by

$$\eta(\gamma) = \prod_{\ell \in \gamma} \widehat{\sigma}_\ell^z \,, \tag{C.4}$$

where $\gamma$ is a closed loop. This is the lattice version of the dual symmetry (also known as the quantum symmetry) in the continuum field theory [115, 54, 102]. See also [120] for related discussions on the lattice. It is easy to check that it commutes with the Gauss law condition (C.3) and therefore, maps within the gauge invariant Hilbert space.

Even though the Hamiltonian is gauge invariant, the operators $Z_v$ and $\widehat{\sigma}_\ell^z$ are not. Next, we would like to write the Hamiltonian in terms of gauge invariant variables. To this end, we define new link variables $\sigma_\ell^x$ and $\sigma_\ell^z$ as follows

$$\sigma_\ell^x = \widehat{\sigma}_\ell^x \,, \qquad \sigma_\ell^z = Z_v \widehat{\sigma}_\ell^z Z_{v'} \,, \tag{C.5}$$

where $\langle v, v' \rangle = \ell$. It is easy to see that these new link variables obey the usual Pauli algebra. In terms of the new link variables, the dual $\mathbb{Z}_2^{(1)}$ symmetry $\eta(\gamma)$ can be written as

$$\eta(\gamma) = \prod_{\ell \in \gamma} \sigma_\ell^z \,. \tag{C.6}$$

Using the Gauss law constraint (C.3), we can write $X$ in terms of the link variables $\sigma_\ell^x$. In terms of these new link variables, we rewrite (C.2) as

$$H_{\text{gauge}} = -J \sum_\ell \sigma_\ell^z - h \sum_v \prod_{\ell \ni v} \sigma_v^x - g \sum_f \prod_{\ell \in f} \sigma_\ell^z \,. \tag{C.7}$$

We will see in the next subsection that this is indeed the Hamiltonian of a $\mathbb{Z}_2$ gauge theory on the dual lattice if we exchange the variables $\sigma_\ell^z \leftrightarrow \sigma_\ell^x$ which can be done using Hadamard operator at every link. Comparing (C.1) with (C.7), we see that gauging maps

$$Z_v Z_{v'} \longleftrightarrow \sigma_{\langle v, v' \rangle}^z \,, \quad X_v \longleftrightarrow \prod_{\ell \ni v} \sigma_\ell^x \,. \tag{C.8}$$

## C.2 Gauging the 1-form symmetry

Conversely, let's start with the $\mathbb{Z}_2$ gauge theory with the following Hamiltonian

$$\widehat{H}_{\text{gauge}} = -J \sum_\ell \sigma_\ell^x - h \sum_f \prod_{\ell \in f} \sigma_\ell^z - g \sum_v \prod_{\ell \ni v} \sigma_\ell^x \,. \tag{C.9}$$

We choose to work on the dual lattice compared to (C.7). We have also performed a Hadamard gate.

The Hamiltonian $\widehat{H}_{\text{gauge}}$ possesses an electric 1-form symmetry $\widehat{\eta}(\widehat{\gamma}) = \prod_{\ell \in \widehat{\gamma}} \sigma_\ell^x$ where $\widehat{\gamma}$ is a curve on the dual lattice and $\ell \in \widehat{\gamma}$ denotes the links $\ell$ that intersect with $\widehat{\gamma}$. In order to

gauge this 1-form symmetry, we introduce 2-form gauge fields on the faces, acted by $X'_f, Z'_f$ and couple it to the Hamiltonian

$$\widehat{H}_{\text{gauge}/\mathbb{Z}_2^{(1)}} = -J \sum_\ell \sigma_\ell^x - h \sum_f X'_f \prod_{\ell \in f} \sigma_\ell^z - g \sum_v \prod_{\ell \ni v} \sigma_\ell^x \,. \tag{C.10}$$

Note that there is no magnetic flux term for the 2-form gauge fields. The Gauss law for the 2-form gauge field is

$$G_\ell = \sigma_\ell^x \prod_{f \ni \ell} Z'_f = Z'_f \sigma_{\langle f,f' \rangle}^x Z'_{f'} = 1 \,. \tag{C.11}$$

We would like to express the Hamiltonian in terms of gauge invariant local variables, but $X'_f$ and $\sigma_\ell^z$ are not. To this end, we define new gauge invariant operators $X_f, Z_f$ on the faces as:

$$X_f = X'_f \prod_{\ell \in f} \sigma_\ell^z \,, \qquad Z_f = Z'_f \,. \tag{C.12}$$

We use the Gauss law to rewrite the Hamiltonian in terms of these new face variables

$$\widehat{H}_{\text{gauge}/\mathbb{Z}_2^{(1)}} = -J \sum_{\langle f,f' \rangle} Z_f Z_{f'} - h \sum_f X_f \,. \tag{C.13}$$

We have dropped the term $g \sum_v \prod_{\ell \ni v} \sigma_\ell^x$ because it is proportional to a constant due to the Gauss law. We have recovered the Ising model on the dual lattice.

In summary, gauging the $\mathbb{Z}_2^{(0)}$ symmetry of the Ising model gives the lattice $\mathbb{Z}_2$ gauge theory. Conversely, gauging the $\mathbb{Z}_2^{(1)}$ symmetry of the lattice gauge theory returns the Ising model.

## C.3   Gauging diagonal $\mathbb{Z}_2^{(0)}$ in Ising$\times$Ising $\to$ (Ising $\times$ Ising) $/\mathbb{Z}_2$

Here, we will derive the Hamiltonian for (Ising $\times$ Ising) $/\mathbb{Z}_2$ in (3.21). We start with the tensor product of two decoupled Ising models on a square lattice with periodic boundary conditions. Its Hamiltonian is

$$H_{\text{Ising}^2} = -h \sum_v X_v - J \sum_{\langle v,v' \rangle} Z_v Z_{v'} - \tilde{h} \sum_v \tilde{X}_v - \tilde{J} \sum_{\langle v,v' \rangle} \tilde{Z}_v \tilde{Z}_{v'} \,. \tag{C.14}$$

For $J = \tilde{J}$ and $h = \tilde{h}$, it has a $D_8$ symmetry generated by

$$U = \prod_v X_v, \quad \tilde{U} = \prod_v \tilde{X}_v, \quad S = \prod_v S_v,$$
$$S_v = \frac{1}{2}\left(1 + X_v\tilde{X}_v + Y_v\tilde{Y}_v + Z_v\tilde{Z}_v\right), \tag{C.15}$$

where $U$ and $\tilde{U}$ are the internal $\mathbb{Z}_2^{(0)}$ and $\tilde{\mathbb{Z}}_2^{(0)}$ symmetries of the respective Ising models. $S$ is the swap operator which swaps $(X_v, Z_v)$ with $(\tilde{X}_v, \tilde{Z}_v)$ and is a symmetry iff $J = \tilde{J}$ and $h = \tilde{h}$. Let's take the diagonal subgroup of $\mathbb{Z}_2^{(0)} \times \tilde{\mathbb{Z}}_2^{(0)}$, generated by $U\tilde{U}$ and gauge it.

To proceed, it is convenient to do a change of basis of the local operators $(X_v, Z_v)$ as follows. We perform a local unitary transformation $\prod_v \mathsf{CNOT}_v$ which implements a CNOT gate $\mathsf{CNOT}_v = \frac{1}{2}(1 + X_v + (1 - X_v)\tilde{Z}_v)$ at every site:

$$X_v \mapsto X_v, \quad Z_v \mapsto Z_v\tilde{Z}_v,$$
$$\tilde{X}_v \mapsto X_v\tilde{X}_v, \quad \tilde{Z}_v \mapsto \tilde{Z}_v. \tag{C.16}$$

In the new basis, the Hamiltonian takes the following form

$$H_{\text{Ising}^2} \mapsto H'_{\text{Ising}^2} = -h\sum_v X_v - J\sum_{\langle v,v'\rangle} Z_v\tilde{Z}_v Z_{v'}\tilde{Z}_{v'} - \tilde{h}\sum_v X_v\tilde{X}_v - \tilde{J}\sum_{\langle v,v'\rangle} \tilde{Z}_v\tilde{Z}_{v'}. \tag{C.17}$$

In terms of the new variables, the original diagonal symmetry $U\tilde{U}$ takes the form $\tilde{U}' = \prod_v \tilde{X}_v$ and it is then straightforward to gauge it.

In order to gauge $\tilde{U}'$, we introduce gauge fields $\widehat{\sigma}_\ell^z, \widehat{\sigma}_\ell^x$ on the links. The Hamiltonian then becomes

$$H_{\frac{\text{Ising} \times \text{Ising}}{\mathbb{Z}_2}} = -h\sum_v X_v - J\sum_{\langle v,v'\rangle} \tilde{Z}_v\widehat{\sigma}_{\langle v,v'\rangle}^z\tilde{Z}_{v'} Z_v Z_{v'} - \tilde{h}\sum_v X_v\tilde{X}_v - \tilde{J}\sum_{\langle v,v'\rangle} \tilde{Z}_v\widehat{\sigma}_{\langle v,v'\rangle}^z\tilde{Z}_{v'} - g\sum_f \prod_{\ell \in f} \widehat{\sigma}_\ell^z. \tag{C.18}$$

Here $\prod_{\ell \in f} \widehat{\sigma}_\ell^z$ is the magnetic flux term. We also impose the Gauss law strictly

$$G_v = \tilde{X}_v \prod_{\ell \ni v} \widehat{\sigma}_\ell^x = 1. \tag{C.19}$$

The operators $\widehat{\sigma}_\ell^z$ and $\tilde{Z}_v$ are not gauge invariant and we would like to express the above Hamiltonian in terms of a new set of gauge invariant operators defined as follows:

$$\sigma_{\langle v,v'\rangle}^z = \tilde{Z}_v\widehat{\sigma}_{\langle v,v'\rangle}^z\tilde{Z}_{v'}, \quad \sigma_{\langle v,v'\rangle}^x = \widehat{\sigma}_{\langle v,v'\rangle}^x, \tag{C.20}$$

whereas $X_v$ and $Z_v$ are unchanged. In terms of these new variables and using the Gauss law

condition, the Hamiltonian becomes (3.21) (up to renaming the parameters)

$$H_{\frac{\text{Ising} \times \text{Ising}}{\mathbb{Z}_2}} = -h \sum_v X_v - J \sum_{\langle v,v' \rangle} Z_v \sigma^z_{\langle v,v' \rangle} Z_{v'} - \tilde{h} \sum_v X_v \prod_{\ell \ni v} \sigma^x_\ell - \tilde{J} \sum_\ell \sigma^z_\ell - g \sum_f \prod_{\ell \in f} \sigma^z_\ell \,.$$

(C.21)

# D   An alternative construction for $\text{Rep}(D_8)$ in 1+1d

In this Appendix, we apply our construction of the non-invertible swap operator in Section 2 to 1+1d. This gives an alternative derivation of the non-invertible $\text{Rep}(D_8)$ symmetry discussed recently in the 1+1d cluster model [28].

## D.1   Non-invertible swap symmetry from gauging

The spatial lattice is a periodic spin chain of $L$ sites labeled by $i \sim i+L$. We start with any Hamiltonian with a $D_8$ symmetry generated by

$$U = \prod_{i=1}^L X_i \,, \qquad \tilde{U} = \prod_{i=1}^L \tilde{X}_i \,,$$

(D.1)

and the swap symmetry

$$S = \prod_{i=1}^L S_i \,, \qquad S_i = \frac{1}{2}\left(1 + X_i \tilde{X}_i + Y_i \tilde{Y}_i + Z_i \tilde{Z}_i\right) \,.$$

(D.2)

They obey the same algebra in (2.11). Each of $U, \tilde{U}, S$ generates a $\mathbb{Z}_2$ subgroup of $D_8 = (\mathbb{Z}_2 \times \tilde{\mathbb{Z}}_2) \rtimes \mathbb{Z}_{2,\text{swap}}$, which we denote by $\mathbb{Z}_2, \tilde{\mathbb{Z}}_2, \mathbb{Z}_{2,\text{swap}}$, respectively. For instance, the quantum Ashkin-Teller model in 1+1d [121–124]:[41]

$$H_{\text{AT}} = -h \sum_{i=1}^L X_i - J \sum_{i=1}^L Z_i Z_{i+1} - \tilde{h} \sum_{i=1}^L \tilde{X}_i - \tilde{J} \sum_{i=1}^L \tilde{Z}_i \tilde{Z}_{i+1} - h' \sum_{i=1}^L X_i \tilde{X}_i - J' \sum_{i=1}^L Z_i \tilde{Z}_i Z_{i+1} \tilde{Z}_{i+1}$$

(D.3)

has a $D_8$ symmetry when $h = \tilde{h}, J = \tilde{J}$. We emphasize that the discussion of symmetries applies more generally to any Hamiltonian with a $D_8$ symmetry.

Next, we gauge the $\tilde{\mathbb{Z}}_2$ symmetry of (D.3) generated by $\tilde{U}$. Concretely, we minimally

---

[41]We thank Sahand Seifnashri for discussions on the non-invertible symmetry in the Ashkin-Teller model.

couple the gauge field $\widehat{\sigma}^z_{i+\frac{1}{2}}$ on the links to $\tilde{Z}_i\tilde{Z}_{i+1}$, and impose the Gauss law

$$\widehat{\sigma}^x_{i-\frac{1}{2}}\tilde{X}_i\widehat{\sigma}^x_{i+\frac{1}{2}} = 1, \tag{D.4}$$

strictly as operator equations. Following [16], we introduce new gauge-invariant variables

$$\sigma^x_{i+\frac{1}{2}} = \widehat{\sigma}^x_{i+\frac{1}{2}}, \quad \sigma^z_{i+\frac{1}{2}} = \tilde{Z}_i\widehat{\sigma}^z_{i+\frac{1}{2}}\tilde{Z}_{i+1}, \tag{D.5}$$

which commute with the Gauss law. Using the Gauss law, we can replace $\tilde{X}_i$ by $\widehat{\sigma}^x_{i-\frac{1}{2}}\widehat{\sigma}^x_{i+\frac{1}{2}}$, and write the gauged Hamiltonian in terms of the new variables. For instance, the Ashkin-Teller Hamiltonian (D.3) becomes

$$\begin{aligned}
H_{\text{noninv}} = & -h\sum_{i=1}^{L} X_i - J\sum_{i=1}^{L} Z_iZ_{i+1} - h\sum_{i=1}^{L}\sigma^x_{i-\frac{1}{2}}\sigma^x_{i+\frac{1}{2}} - J\sum_{i=1}^{L}\sigma^z_{i-\frac{1}{2}} \\
& -h'\sum_{i=1}^{L}\sigma^x_{i-\frac{1}{2}}X_i\sigma^x_{i+\frac{1}{2}} - J'\sum_{i=1}^{L}Z_i\sigma^z_{i+\frac{1}{2}}Z_{i+1}.
\end{aligned} \tag{D.6}$$

Since $U$ is not gauged, it survives as the symmetry for the new model. Moreover, there is a dual $\widehat{\mathbb{Z}}_2$ symmetry generated by[42]

$$\eta = \prod_{i=1}^{L}\sigma^z_{i-\frac{1}{2}}. \tag{D.7}$$

How about the swap symmetry $S$?

For the $\tilde{X}_i$ factors in the swap operator $S$, we can simply replace it with $\sigma^x_{i-\frac{1}{2}}\sigma^x_{i+\frac{1}{2}}$ using the Gauss law. But it is less clear what to do with the $\tilde{Z}_i$ factors, which are charged under the gauged symmetry $\tilde{U}$. Intuitively, the $\tilde{\mathbb{Z}}_2$-charged operator should be attached to a Wilson line after gauging. Following this intuition, to each $\tilde{Z}_i$, we attach a string of $\widehat{\sigma}^z$'s which starts at an arbitrarily chosen reference point $i = 0$, and end at site $i$. We can then express it in terms of the gauge-invariant variables $\sigma^z$'s. Combining the above steps together, the swap operator (D.2) becomes

$$\mathsf{S} = \prod_{i=1}^{L}\mathsf{S}_i, \quad \mathsf{S}_i = \frac{1}{2}\left(1 + \sigma^x_{i-\frac{1}{2}}X_i\sigma^x_{i+\frac{1}{2}} + Z_i\prod_{j=1}^{i}\sigma^z_{j-\frac{1}{2}}(1 - \sigma^x_{i-\frac{1}{2}}X_i\sigma^x_{i+\frac{1}{2}})\right). \tag{D.8}$$

Note that $\mathsf{S}_i$ is unitary for $i = 1, 2, \cdots, L - 1$, but $\mathsf{S}_L$ is not. Hence $\mathsf{S}$ is no longer a unitary

---

[42]In Section 2, the dual symmetry $\eta$ in 2+1d is a 1-form symmetry, while it is a 0-form symmetry here in 1+1d.

operator. One can verify that

$$Z_i Z_{i+1} \mathsf{S} = \mathsf{S}\sigma^z_{i+\frac{1}{2}}, \qquad X_i \mathsf{S} = \mathsf{S}\sigma^x_{i-\frac{1}{2}}\sigma^x_{i+\frac{1}{2}},$$
$$\mathsf{S}Z_i Z_{i+1} = \sigma^z_{i+\frac{1}{2}}\mathsf{S}, \qquad \mathsf{S}X_i = \sigma^x_{i-\frac{1}{2}}\sigma^x_{i+\frac{1}{2}}\mathsf{S}, \qquad i = 1, ..., L-1. \tag{D.9}$$

It exchanges the first and third terms, as well as the second and fourth terms of $H_{\mathrm{noninv}}$, are exchanged under $\mathsf{S}$, on most of the sites/links. But for the operator around site $L$, we find some non-local operators involving $\eta$ and $U$ on the righthand side:

$$\mathsf{S}Z_L Z_1 = \sigma^z_{\frac{1}{2}}\eta\mathsf{S}, \qquad Z_L Z_1 \mathsf{S} = \mathsf{S}\sigma^z_{\frac{1}{2}}\eta,$$
$$\mathsf{S}X_L = \sigma^x_{L-\frac{1}{2}}\sigma^x_{\frac{1}{2}}\mathsf{S}U, \qquad X_L \mathsf{S} = \mathsf{S}\sigma^x_{L-\frac{1}{2}}\sigma^x_{\frac{1}{2}}. \tag{D.10}$$

This issue can be cured by multiplying $\mathsf{S}$ by projectors $(1+U)(1+\eta)$. We therefore get the non-invertible swap operator

$$\mathsf{D} = \frac{1}{2}\mathsf{S}(1+U)(1+\eta), \tag{D.11}$$

which satisfies the fusion rule

$$\mathsf{D}^2 = (1+U)(1+\eta), \quad \mathsf{D}\eta = \eta\mathsf{D} = \mathsf{D}, \quad \mathsf{D}U = U\mathsf{D} = \mathsf{D}. \tag{D.12}$$

The non-invertible swap operator then acts correctly on the terms in the Hamiltonian for every site:

$$\mathsf{D}Z_i Z_{i+1} = \sigma^z_{i+\frac{1}{2}}\mathsf{D}, \qquad \mathsf{D}X_i = \sigma^x_{i-\frac{1}{2}}\sigma^x_{i+\frac{1}{2}}\mathsf{D},$$
$$Z_i Z_{i+1}\mathsf{D} = \mathsf{D}\sigma^z_{i+\frac{1}{2}}, \qquad X_i\mathsf{D} = \mathsf{D}\sigma^x_{i-\frac{1}{2}}\sigma^x_{i+\frac{1}{2}}. \tag{D.13}$$

**Field theory interpretation**

The above results again admit a simple field theory explanation. We start with a theory $\mathcal{T}'$ with a $D_8$ symmetry. Denote the background field for $\mathbb{Z}_2 \times \tilde{\mathbb{Z}}_2$ generated by $U$ and $\tilde{U}$ by $A, \tilde{A}$, respectively. The partition function is $\mathcal{Z}_{\mathcal{T}'}[M_2; A, \tilde{A}]$. Invariance under the swap symmetry, generated by $S$, means $\mathcal{Z}_{\mathcal{T}'}[M_2; A, \tilde{A}] = \mathcal{Z}_{\mathcal{T}'}[M_2; \tilde{A}, A]$. Now we gauge $\tilde{\mathbb{Z}}_2$ by promoting the background field $\tilde{A}$ to $a$, and denote the resulting theory as $\mathcal{T}$, whose partition function is[43]

$$\mathcal{Z}_{\mathcal{T}}[M_2; A, B] = \sum_a \mathcal{Z}_{\mathcal{T}'}[M_2; A, a](-1)^{\int_{M_2} a\cup B}, \tag{D.14}$$

---

[43] For simplicity, we omit the overall normalization factors in the partition functions.

where $B$ is the background field for the dual $\widehat{\mathbb{Z}}_2$ symmetry $\eta$. It follows that the theory $\mathcal{T}$ is invariant under gauging $\mathbb{Z}_2 \times \widehat{\mathbb{Z}}_2$:

$$
\begin{aligned}
\sum_{a,b} \mathcal{Z}_\mathcal{T}[M_2; a, b](-1)^{\int_{M_2} a \cup B + b \cup A} &= \sum_{a,b,a'} \mathcal{Z}_{\mathcal{T}'}[M_2; a, a'](-1)^{\int_{M_2} a' \cup b + a \cup B + b \cup A} \\
&= \sum_a \mathcal{Z}_{\mathcal{T}'}[M_2; a, A](-1)^{\int_{M_2} a \cup B} \\
&= \sum_a \mathcal{Z}_{\mathcal{T}'}[M_2; A, a](-1)^{\int_{M_2} a \cup B} \\
&= \mathcal{Z}_\mathcal{T}[M_2; A, B] \,.
\end{aligned}
\tag{D.15}
$$

We have used the swap symmetry $S$ of $\mathcal{T}'$ in the third equality. Therefore, the theory $\mathcal{T}$ is self-dual under gauging $\mathbb{Z}_2 \times \widehat{\mathbb{Z}}_2$. The half gauging construction then leads to a non-invertible symmetry in $\mathcal{T}$, which comes from the invertible swap symmetry $S$ of $\mathcal{T}'$, and the resulting non-invertible duality defect moreover satisfies fusion rules analogous to (D.12).

## D.2 $\mathrm{Rep}(D_8)$ and the Kramers-Wannier symmetry

The non-invertible swap operator $\mathsf{D}$ admits another expression

$$
\mathsf{D} = T \mathsf{D}_{\mathrm{site}} \mathsf{D}_{\mathrm{link}} \,.
\tag{D.16}
$$

Below we explain each factor. First, $T$ is the translation by half of a unit cell (conjugated by the Hadamard gate on the link variables):

$$
T = \left( \prod_{i=1}^{L-1} R_{i-\frac{1}{2},i} R_{i,i+\frac{1}{2}} \right) R_{L-\frac{1}{2},L} \,,
\tag{D.17}
$$

with

$$
R_{i,i+\frac{1}{2}} = \frac{1}{2} \left( 1 + X_i \sigma_{i+\frac{1}{2}}^z + Z_i \sigma_{i+\frac{1}{2}}^x - Y_i \sigma_{i+\frac{1}{2}}^y \right) \,, \quad R_{i-\frac{1}{2},i} = \frac{1}{2} \left( 1 + \sigma_{i-\frac{1}{2}}^z X_i + \sigma_{i-\frac{1}{2}}^x Z_i - \sigma_{i-\frac{1}{2}}^y Y_i \right) \,.
\tag{D.18}
$$

It acts on the local operators as

$$
\begin{aligned}
T Z_i &= \sigma_{i+\frac{1}{2}}^x T \,, \quad & T X_i &= \sigma_{i+\frac{1}{2}}^z T \,, \quad & T Y_i &= -\sigma_{i+\frac{1}{2}}^y T \,, \\
Z_i T &= T \sigma_{i+\frac{1}{2}}^x \,, \quad & X_i T &= T \sigma_{i+\frac{1}{2}}^z \,, \quad & Y_i T &= -T \sigma_{i+\frac{1}{2}}^y \,.
\end{aligned}
\tag{D.19}
$$

Second, $\mathsf{D}_{\mathrm{site}}$ is the Kramers-Wannier duality operator acting on the sites [15, 16]

$$\mathsf{D}_{\mathrm{site}} = \frac{e^{\frac{2\pi i L}{8}}}{2^L}(1+U)(1-iX_L)(1-iZ_LZ_{L-1})(1-iX_{L-1})...(1-iZ_2Z_1)(1-iX_1) \quad \text{(D.20)}$$

which acts on the local operators as

$$\mathsf{D}_{\mathrm{site}}X_{i+1} = Z_iZ_{i+1}\mathsf{D}_{\mathrm{site}}, \qquad \mathsf{D}_{\mathrm{site}}Z_iZ_{i+1} = X_i\mathsf{D}_{\mathrm{site}}. \quad \text{(D.21)}$$

Third, $\mathsf{D}_{\mathrm{link}}$ operator is the Kramers-Wannier duality operator (up to a Hadamard gate) acting on the link variables,

$$\mathsf{D}_{\mathrm{link}} = \frac{e^{\frac{2\pi i L}{8}}}{2^L}(1+\eta)(1-i\sigma^z_{L-\frac{1}{2}})(1-i\sigma^x_{L-\frac{1}{2}}\sigma^x_{L-\frac{3}{2}})(1-i\sigma^z_{L-\frac{3}{2}})...(1-i\sigma^x_{\frac{3}{2}}\sigma^x_{\frac{1}{2}})(1-i\sigma^z_{\frac{1}{2}}) \quad \text{(D.22)}$$

which acts on the local operators as

$$\mathsf{D}_{\mathrm{link}}\sigma^z_{i+\frac{1}{2}} = \sigma^x_{i-\frac{1}{2}}\sigma^x_{i+\frac{1}{2}}\mathsf{D}_{\mathrm{link}}, \qquad \mathsf{D}_{\mathrm{link}}\sigma^x_{i-\frac{1}{2}}\sigma^x_{i+\frac{1}{2}} = \sigma^z_{i-\frac{1}{2}}\mathsf{D}_{\mathrm{link}}. \quad \text{(D.23)}$$

The presentation (D.16) of the non-invertible swap operator matches the one in [28, Eq. (5)]. This operator $\mathsf{D}$, together with the $\mathbb{Z}_2 \times \widehat{\mathbb{Z}}_2$ symmetry generated by $U, \eta$, form a $\mathrm{Rep}(D_8)$ fusion category as shown in [28].

This new way of presentation has the advantage that the operator is now a product of local factors times global symmetry projectors, while in our original presentation (D.11), each factor within $\mathsf{S}$, i.e. $\mathsf{S}_i$, is non-local.

We note that a Hamiltonian may not be invariant under $\mathsf{D}_{\mathrm{site}}$ and $\mathsf{D}_{\mathrm{link}}$ separately, but is only invariant under the combination (D.16). For instance, the Hamiltonian (D.6) commutes with $\mathsf{D}$ for any $h, J, h', J'$, while it commutes with $\mathsf{D}_{\mathrm{link}}$ (and separately $\mathsf{D}_{\mathrm{site}}$) only when $(h, h') = (J, J')$.

**Field theory interpretation**

There is a simple field theory explanation for the expression (D.16). Since the operator $\mathsf{D}_{\mathrm{site}}$ implements the gauging of the $\mathbb{Z}_2$ symmetry on the sites, its action amounts to promoting $A$ in $\mathcal{Z}_{\mathcal{T}}[M_2; A, B]$ to a dynamical field $a$. Similarly, since the operator $\mathsf{D}_{\mathrm{link}}$ implements the gauging of the $\widehat{\mathbb{Z}}_2$ symmetry on the links, its action amounts to promoting $B$ in $\mathcal{Z}_{\mathcal{T}}[M_2; A, B]$ to a dynamical field $b$. We then couple the gauged theory to the background gauge fields $\widehat{A}, \widehat{B}$ for the two dual symmetries. Explicitly, the lattice operator $\mathsf{D}_{\mathrm{site}}\mathsf{D}_{\mathrm{link}}$ corresponds to

gauging the $\mathbb{Z}_2 \times \widehat{\mathbb{Z}}_2$ symmetry of $\mathcal{T}$ in the continuum as follows:

$$
\begin{aligned}
\sum_{a,b} \mathcal{Z}_{\mathcal{T}}[M_2; a, b](-1)^{\int_{M_2} a \cup \widehat{A} + b \cup \widehat{B}} &= \sum_{a,b,b'} \mathcal{Z}_{\mathcal{T}'}[M_2; a, b'](-1)^{\int_{M_2} b' \cup b + a \cup \widehat{A} + b \cup \widehat{B}} \\
&= \sum_{a} \mathcal{Z}_{\mathcal{T}'}[M_2; a, \widehat{B}](-1)^{\int_{M_2} a \cup \widehat{A}} = \mathcal{Z}_{\mathcal{T}}[M_2; \widehat{B}, \widehat{A}].
\end{aligned}
\tag{D.24}
$$

We find that the gauged partition function has the two background gauge fields exchanged, which is implemented by $T$ on the lattice. This gives the continuum explanation as to why $T \mathsf{D}_{\mathrm{site}} \mathsf{D}_{\mathrm{link}}$ leaves the theory $\mathcal{T}$ invariant.

# E  More on the non-invertible swap operator in 2+1d

In this Appendix we provide more details on the non-invertible operator $\mathsf{D}$ in Section 2.

## E.1  Action on local operators

We begin with the operator $\mathsf{S}$ given by[44]

$$
\mathsf{S} = \left( \prod_{v \neq v_0} \mathsf{S}_v \right) \mathsf{S}_{v_0}, \qquad \mathsf{S}_v = \frac{1}{2} \left( 1 + G_v + W_{v_0, v} Z_v (1 - G_v) \right).
\tag{E.1}
$$

We have

$$
\mathsf{S}_v \mathsf{S}_{v'} = \mathsf{S}_{v'} \mathsf{S}_v \ , \forall\, v, v' \neq v_0, \qquad \mathsf{S}_v^2 = 1 \ , \forall\, v \neq v_0, \qquad \mathsf{S}_{v_0}^2 = \mathsf{S}_{v_0}.
\tag{E.2}
$$

In particular, $\mathsf{S}_v$ for $v \neq v_0$ is an invertible operator whereas $\mathsf{S}_{v_0}$ is non-invertible. We also note the commutation relation of $G_v$ with $\mathsf{S}_v$,

$$
\mathsf{S}_v G_{v'} = \begin{cases} G_{v'} \mathsf{S}_v, & \text{if } v' \neq v_0, \\ G_{v'} \mathsf{S}_v G_v, & \text{if } v' = v_0. \end{cases}
\tag{E.3}
$$

---

[44]Recall that the expression for $\mathsf{S}$ was motivated from that of the swap operator $S$ in (2.10). There are two equally valid choices of $\mathsf{S}_v$ that can come from $S_v$, i.e., $\mathsf{S}_v = \frac{1}{2} \left( 1 + G_v + W_{v_0, v} Z_v (1 - G_v) \right)$ or $\tilde{\mathsf{S}}_v = \frac{1}{2} \left( 1 + G_v + (1 - G_v) W_{v_0, v} Z_v \right)$. They are equal for all $v$ except for $v = v_0$. Both choices can be used to write the non-invertible swap operator $\mathsf{D}$. In the latter case, the non-invertible swap operator takes the form $\mathsf{D} = \frac{1}{2} \mathsf{C}_\eta \mathsf{C}_U \tilde{\mathsf{S}}_{v_0} \left( \prod_{v \neq v_0} \tilde{\mathsf{S}}_v \right)$.

**Action on $X_v$ and $\prod_{\ell \ni v} \sigma_\ell^x$**

We first discuss how $\mathsf{S}$ acts on $X_v$ for $v \neq v_0$. Note that

$$\mathsf{S}_v X_{v'} = \begin{cases} X_{v'} \mathsf{S}_v \,, & \text{if } v \neq v' \,, \\ X_{v'} \mathsf{S}_v G_v \,, & \text{if } v = v' \,. \end{cases} \tag{E.4}$$

Then, using (E.3) and (E.4), we have

$$\mathsf{S} X_v = \left( \prod_{\ell \ni v} \sigma_\ell^x \right) \mathsf{S} \,, \quad \forall\, v \neq v_0 \,. \tag{E.5}$$

This gives us the first equation of (2.17), and the second equation can be derived similarly.

In order to find the action of $\mathsf{S}$ on $X_{v_0}$, we use the fact that

$$U\mathsf{S} = \mathsf{S} \,. \tag{E.6}$$

This can be seen as follows. First, we have $U\mathsf{S}_v = \mathsf{S}_v G_v U$, and

$$U\mathsf{S} = \left( \prod_{v \neq v_0} \mathsf{S}_v G_v \right) \mathsf{S}_{v_0} G_{v_0} U = \left( \prod_{v \neq v_0} \mathsf{S}_v \right) \mathsf{S}_{v_0} \left( \prod_{v'} G_{v'} \right) U = \mathsf{S} \,. \tag{E.7}$$

where we have used $\prod_{v'} G_{v'} = \prod_{v'} X_{v'} = U$ and $U^2 = 1$ in the last equality.

Using (E.4), we have

$$\mathsf{S} X_{v_0} = X_{v_0} \mathsf{S} G_{v_0} \,. \tag{E.8}$$

Now, on the right-hand side of the above equation, we can freely insert $U = \prod_v G_v$ to the left of $\mathsf{S}$ because of (E.6). It gives

$$\mathsf{S} X_{v_0} = X_{v_0} \left( \prod_v G_v \right) \mathsf{S} G_{v_0} = X_{v_0} G_{v_0} \mathsf{S} \left( \prod_v G_v \right) = \left( \prod_{\ell \ni v_0} \sigma_\ell^x \right) \mathsf{S} U \,. \tag{E.9}$$

where in the second equality we separated $G_{v_0}$ from the product of $G_v$'s, and then pushed the remaining $G_v$'s to the right to get $\prod_v G_v = U$. This proves the first equation of (2.18), and the second equation can be derived similarly.

From above, it follows that the non-invertible symmetry operator $\mathsf{D} = \frac{1}{2}\mathsf{SC} = \frac{1}{2}\mathsf{SC}_U \mathsf{C}_\eta$

acts on $X_v$ and $\prod_{\ell \ni v} \sigma_\ell^x$ as desired:

$$\mathsf{D}X_v = \left(\prod_{\ell \ni v} \sigma_\ell^x\right)\mathsf{D}, \quad \mathsf{D}\left(\prod_{\ell \ni v} \sigma_\ell^x\right) = X_v\mathsf{D}. \tag{E.10}$$

This is because $\mathsf{C}_U = 1 + U$ absorbs $U$, and the condensation operator $\mathsf{C}_\eta$ commutes with both $X_v$ and $\prod_{\ell \ni v} \sigma_\ell^x$.

**Action on $Z_v Z_{v'}$ and $\sigma_{\langle v, v'\rangle}^z$**

First note that

$$\mathsf{S}_v Z_{v'} = \begin{cases} Z_{v'}\mathsf{S}_v, & \text{if } v \neq v', \\ W_{v_0,v}\mathsf{S}_v, & \text{if } v = v', \end{cases} \tag{E.11}$$

and also

$$\begin{aligned}
\mathsf{S}_{v'}W_{v_0,v} &= W_{v_0,v}\mathsf{S}_{v'} \quad \text{unless } v' = v_0 \text{ or } v' = v, \\
\mathsf{S}_v W_{v_0,v} &= Z_v\mathsf{S}_v, \\
\mathsf{S}_{v_0}W_{v_0,v} &= Z_{v_0}W_{v_0,v}\mathsf{S}_{v_0}.
\end{aligned} \tag{E.12}$$

Using (E.11) and (E.12), we find

$$\mathsf{S}Z_v = W_{v_0,v}\mathsf{S}, \quad \forall\, v. \tag{E.13}$$

This implies that

$$\mathsf{S}\mathsf{C}_U Z_v Z_{v'} = \mathsf{S}Z_v Z_{v'}\mathsf{C}_U = W_{v_0,v}W_{v_0,v'}\mathsf{S}\mathsf{C}_U, \tag{E.14}$$

where we first used the fact that $Z_v Z_{v'}$ commutes with $\mathsf{C}_U$ and then we used (E.13). This gives us the first equation of (2.20). The second equation of (2.20) can be derived similarly.

Combined with (2.23), we see that the non-invertible operator $\mathsf{D} = \frac{1}{2}\mathsf{S}\mathsf{C}_U\mathsf{C}_\eta = \frac{1}{2}\mathsf{S}\mathsf{C}$ acts on $Z_v Z_{v'}$ and $\sigma_{\langle v, v'\rangle}^z$ as desired:

$$\mathsf{D}Z_v Z_{v'} = \sigma_{\langle v, v'\rangle}^z\mathsf{D}, \quad \mathsf{D}\sigma_{\langle v, v'\rangle}^z = Z_v Z_{v'}\mathsf{D}, \tag{E.15}$$

from which (2.8) also follows.

## E.2 Translation invariance of D

Below we explicitly show that the non-invertible symmetry operator $\mathsf{D}$ does not depend on the choice of the reference point $v_0$. First, we define the following projectors

$$G_v^{\pm} = \frac{1}{2}\left(1 \pm G_v\right) . \tag{E.16}$$

Let us label the vertices on the periodic square lattice as $v_{A-1}, v_{A-2}, ..., v_0$ where $A$ is the total number of vertices. We rewrite the operator $\mathsf{S}$ in (E.1) as

$$\mathsf{S} = \left(G_{v_{A-1}}^+ + W_{v_0,v_{A-1}} Z_{v_{A-1}} G_{v_{A-1}}^-\right) \left(G_{v_{A-2}}^+ + W_{v_0,v_{A-2}} Z_{v_{A-2}} G_{v_{A-2}}^-\right) \cdots \left(G_{v_0}^+ + W_{v_0,v_0} Z_{v_0} G_{v_0}^-\right),$$
$$\tag{E.17}$$

where $W_{v_0,v_0} = 1$. We can expand the product above and collect the terms in which $W_{v_0,v_i} Z_{v_i} G_{v_i}^-$ appears $k$ number of times and $G_{v_i}^+$ appears $A - k$ number of times, for each $k \in \{0, 1, ..., A\}$. That is, we can write

$$\mathsf{S} = \sum_{k=0}^{A} \mathsf{S}^{(k)} , \tag{E.18}$$

where $\mathsf{S}^{(k)}$ is defined as

$$\mathsf{S}^{(k)} = \sum_{i_1 > i_2 > \cdots > i_k} \mathsf{S}^{(k)}_{i_1, i_2, ..., i_k} ,$$

$$\mathsf{S}^{(k)}_{i_1, i_2, ..., i_k} = \left(\prod_{m \in \{i_1, i_2, ..., i_k\}} Z_{v_m} W_{v_0, v_m} G_{v_m}^-\right) \prod_{l \notin \{i_1, i_2, ..., i_k\}} G_{v_l}^+ . \tag{E.19}$$

Here, $i_1, i_2, ..., i_k \in \{0, 1, 2, ..., A - 1\}$ and $i_1 > i_2 > \cdots > i_k$ to avoid overcounting and also to respect the ordering of the product chosen in (E.17). In addition, using the fact that $G_v$ commutes with $W_{v_0,v'} Z_{v'}$ for all $v, v'$ except when $v = v' = v_0$, we brought all $G_v^+$'s to the right in the expression of $\mathsf{S}^{(k)}_{i_1, i_2, ..., i_k}$.

Now, note that $U\mathsf{D} = \mathsf{D}$ due to (E.6), and hence

$$\mathsf{D} = \frac{1}{2}\mathsf{C}_U\mathsf{D} = \frac{1}{4}\mathsf{C}_U\mathsf{S}\mathsf{C}_U\mathsf{C}_\eta . \tag{E.20}$$

Since we have $\mathsf{C}_U = 1 + U$ multiplying $\mathsf{S}$ from both sides, in the expression (E.18) of $\mathsf{S}$, only the terms which are even under $U$ contribute, namely the operators $\mathsf{S}^{(2k)}$ which contain $Z_v$ even number of times.

Therefore, we have

$$\mathsf{D} = \frac{1}{2} \sum_{k=0}^{\lfloor \frac{A}{2} \rfloor} \mathsf{S}^{(2k)} \mathsf{C}_U \mathsf{C}_\eta \,. \tag{E.21}$$

Recall that the potential dependence on the reference point $v_0$ of $\mathsf{D}$ comes from the presence of Wilson line operators $W_{v_0,v}$. In the expression of $\mathsf{S}^{(2k)}$ in (E.19), Wilson line operators appear $2k$ number of times. In (E.21), we may group the Wilson line operators in each $\mathsf{S}^{(2k)}$ into $k$ pairs and use

$$W_{v_0,v} W_{v_0,v'} \mathsf{C}_\eta = W_{v,v'} \mathsf{C}_\eta \ , \forall \, v, v' \,. \tag{E.22}$$

Explicitly, using (E.19), (E.21), and (E.22), we can write the operator $\mathsf{D}$ as

$$\mathsf{D} = \frac{1}{2} \sum_{k=0}^{\lfloor \frac{A}{2} \rfloor} \left[ \left( \prod_{j=1}^{k} Z_{v_{i_j}} W_{v_{i_j}, v_{i_{j+k}}} Z_{v_{i_{j+k}}} G_{v_{i_j}}^- G_{v_{i_{j+k}}}^- \right) \left( \prod_{l \notin \{i_1, i_2, \dots, i_{2k}\}} G_{v_l}^+ \right) \right] \mathsf{C}_U \mathsf{C}_\eta \,, \tag{E.23}$$

and we see that $\mathsf{D}$ is manifestly independent of the choice of $v_0$.

## E.3 Hermiticity of $\mathsf{D}$

From the expression (E.23) for the operator $\mathsf{D}$, it immediately follows that

$$\mathsf{D} = \mathsf{D}^\dagger \,. \tag{E.24}$$

This is because the operators $Z_v W_{v,v'} Z_{v'}$, $G_v^\pm$, $\mathsf{C}_U$ and $\mathsf{C}_\eta$ are all hermitian and mutually commute with each other.

## E.4 Derivation of the operator algebra

Here we derive the operator algebras given in (2.36) and (2.37). First, using the fact that $\mathsf{D}$ is hermitian, we have

$$\begin{aligned} \mathsf{D}^2 = \mathsf{D}^\dagger \mathsf{D} &= \frac{1}{4} \mathsf{C} \mathsf{S}^\dagger \mathsf{S} \mathsf{C} = \frac{1}{4} \mathsf{C} \mathsf{S}_{v_0}^\dagger \left( \prod_{v \neq v_0} \mathsf{S}_v^\dagger \right) \left( \prod_{v \neq v_0} \mathsf{S}_v \right) \mathsf{S}_{v_0} \mathsf{C} \\ &= \frac{1}{4} \mathsf{C} (1 + Z_{v_0}) \mathsf{C} = \frac{1}{4} \mathsf{C}^2 = \mathsf{C} \,. \end{aligned} \tag{E.25}$$

We have used $\mathsf{S}_v^\dagger \mathsf{S}_v = 1$ for all $v \neq v_0$, $\mathsf{S}_{v_0}^\dagger \mathsf{S}_{v_0} = 1 + Z_{v_0}$, and $(1+U) Z_{v_0} (1+U) = 0$. Moreover, $\mathsf{S}_v$ and $\mathsf{S}_{v'}$ commute with each other for all $v, v' \neq v_0$. Finally, the last equality of (E.25)

follows from

$$C^2 = 4C \,, \tag{E.26}$$

which can be seen immediately from (2.25) and $C = (1+U)C_\eta$. Furthermore, (E.26) implies that

$$DC = CD = 4D \,, \tag{E.27}$$

since $D = \frac{1}{2}SC$.[45] Finally, we have

$$DU = UD = D \,, \quad D\eta(\gamma) = \eta(\gamma)D = D \,. \tag{E.28}$$

The fact that $UD = D$ follows from (E.6), and other relations are straightforward to verify.

# F   Anomalies at the SPT transition point

## F.1   SPT entangler

On the lattice, the two $\mathbb{Z}_2^{(0)} \times \mathbb{Z}_2^{(1)}$ SPT phases are described by the $\mathbb{Z}_2^{(0)} \times \mathbb{Z}_2^{(1)}$-symmetric product state $|\text{prod.}\rangle$ and the cluster state $|\text{cluster}\rangle$, which are the unique ground states of the exactly solvable Hamiltonians $H_{\text{trivial}}$ and $H_{\text{cluster}}$, respectively. These Hamiltonians are[46]

$$
\begin{aligned}
H_{\text{trivial}} &= -\sum_v X_v - \sum_\ell \sigma_\ell^z \,, \\
H_{\text{cluster}} &= -\sum_v X_v \prod_{\ell \ni v} \sigma_\ell^x - \sum_{\langle v,v' \rangle} Z_v \sigma_{\langle v,v' \rangle}^z Z_{v'}.
\end{aligned}
\tag{F.1}
$$

The two Hamiltonians are related by the entangler $V$ defined in (3.2),

$$V H_{\text{trivial}} V^{-1} = H_{\text{cluster}} \,. \tag{F.2}$$

Hence, their ground states are also related by the entangler,

$$V|\text{prod.}\rangle = |\text{cluster}\rangle \,. \tag{F.3}$$

In [125], two short-range entangled states are defined to describe the same SPT phase if they are related by a *locally symmetric finite depth unitary operator*. Note that in (F.3), the entangler $V$ is *not* a locally symmetric finite depth unitary operator. This is because

---

[45]Using the fact that $D$ and $C$ are hermitian, we have $CD = (DC)^\dagger = 4D^\dagger = 4D$.

[46]Since both Hamiltonians are commuting Pauli Hamiltonians, the magnitudes of the coupling constants do not affect the phase diagram. We set these coupling constants to be 1 for brevity.

despite the global $V$ commutes with both $U$ and $\eta(\gamma)$, the local factor $V_v$ does not. This is consistent with the fact that $|\text{prod.}\rangle$ and $|\text{cluster}\rangle$ describe different SPT phases. While the notion of invertible SPT phases is generally relative, on the lattice with a tensor product Hilbert space, it is natural to declare $|\text{prod.}\rangle$ as the "trivial" SPT state, and $|\text{cluster}\rangle$ as the "non-trivial" SPT state. However, such a distinction is not universal (see [27, 28]).

Having chosen $|\text{prod.}\rangle$ to be the "trivial" SPT state, acting on an arbitrary state $|\psi\rangle$ with the entangler $V$ amounts to stacking a cluster state. To see this, stacking $|\text{cluster}\rangle$ to $|\psi\rangle$ gives $|\psi\rangle \otimes |\text{cluster}\rangle$, where the Hilbert space is doubled. We then apply the diagonal entangler $V \otimes V$ to the this state to find

$$V \otimes V \left( |\psi\rangle \otimes |\text{cluster}\rangle \right) = (V|\psi\rangle) \otimes |\text{prod.}\rangle \tag{F.4}$$

which is identical to $V|\psi\rangle$ by striping off the product state. The same argument works for Hamiltonian as well — stacking $H$ with an $\mathbb{Z}_2^{(0)} \times \mathbb{Z}_2^{(1)}$ SPT can be achieved by conjugating with the entangler $V$.

## F.2 Anomaly eats an SPT state

Suppose a Hamiltonian has the type III anomaly discussed in Section 3, then it must in particular commute with the entangler $V$. This then implies that it is invariant under stacking the cluster SPT state, i.e., it "eats" the SPT state. An example of such a Hamiltonian is (3.1) with $h = \tilde{J} = 1, J = \tilde{h} = g = 0$, i.e., $H = H_{\text{trivial}} + H_{\text{cluster}}$, which can be viewed as the transition point between the two SPT phases. It commutes with $V$, which exchanges exchange $H_{\text{trivial}}$ with $H_{\text{cluster}}$. See [28] for the corresponding discussion in $1 + 1d$.

The field theory interpretation of this phenomenon is that any QFT $\mathcal{T}''$ with a type III anomaly of $\mathbb{Z}_2^{(0)} \times \mathbb{Z}_2^{(1)} \times \mathbb{Z}_{2,\text{swap}}^{(0)}$ must be invariant under stacking an invertible field theory. More precisely, following the notations in Appendix B, we expect

$$\mathcal{Z}_{\mathcal{T}''}[M_3; A, C, B] = \mathcal{Z}_{\mathcal{T}''}[M_3; A, C, B](-1)^{\int_{M_3} B \cup A} . \tag{F.5}$$

Here, $(-1)^{\int_{M_3} B \cup A}$ is the partition function of the 2+1d invertible field theory representing the relative difference between the $\mathbb{Z}_2^{(0)} \times \mathbb{Z}_2^{(1)}$ SPT phases.

Equation (F.5) can be shown as follows. The type III anomaly implies that, under the background gauge transformation $C \to C + \delta\alpha$, the partition function of $\mathcal{T}''$ transforms as

$$\mathcal{Z}_{\mathcal{T}''}[M_3; A, C + \delta\alpha, B] = \mathcal{Z}_{\mathcal{T}''}[M_3; A, C, B](-1)^{\int_{M_3} B \cup \alpha \cup A} . \tag{F.6}$$

Here, $\alpha$ is an arbitrary $\mathbb{Z}_2$-valued 0-cochain. Now, consider the special case where $\alpha$ is a

constant everywhere in spacetime, and takes the value $1 \in \mathbb{Z}_2 = \{0, 1\}$. In this case, $\delta\alpha = 0$, and (F.6) reduces to the desired equation (F.5).

An interesting consequence of (F.5) is that the partition function of $\mathcal{T}''$ vanishes on background configurations where the cluster SPT partition function is nontrivial, i.e., whenever $(-1)^{\int_{M_3} B \cup A} = -1$.

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
