# Peer review of "Non-invertible and higher-form symmetries in 2+1d lattice gauge theories"

_SciPost Physics_

## Round 1 · Referee Report · Anonymous (Referee 1) · 2024-7-25

Strengths

  1. Thorough investigation of generalized symmetries, including non-invertible ones, of lattice $\mathbb{Z}_2$ gauge theories in 2+1 dimensions.
  2. Concrete lattice models of two SPT phases with non-invertible fusion 2-category symmetry $2\text{Rep}((\mathbb{Z}_2^{(1)} \times \mathbb{Z}_2^{(1)}) \rtimes \mathbb{Z}_2^{(0)})$.
  3. Precise formulation of a generalized Kennedy-Tasaki transformation (i.e., a twisted gauging) in the context of 2+1d lattice $\mathbb{Z}_2$ gauge theories.
  4. Detailed derivations of the results are provided in appendices.
  5. The manuscript is clearly written.

Report

This paper revealed non-invertible and invertible symmetries of 2+1d lattice $\mathbb{Z}_2$ gauge theories coupled to matter degrees of freedom. The model studied in the paper generally has a $\mathbb{Z}_2^{(0)} \times \mathbb{Z}_2^{(1)}$ symmetry, where superscripts 0 and 1 denote the form degree. When the model is self-dual under gauging this symmetry, it enjoys a larger symmetry extended by the non-invertible duality operator $\mathsf{D}$ that implements the gauging. The authors found an explicit expression of the duality operator $\mathsf{D}$ on the lattice and derived the algebra of symmetry operators involving $\mathsf{D}$. The operator algebra matches that in the continuum, suggesting that the total symmetry is described by a fusion 2-category $2\text{Rep}((\mathbb{Z}_2^{(1)} \times \mathbb{Z}_2^{(1)}) \rtimes \mathbb{Z}_2^{(0)})$. On the other hand, when the model is invariant under stacking the non-trivial $\mathbb{Z}_2^{(0)} \times \mathbb{Z}_2^{(1)}$ SPT phase, the total symmetry of the model becomes $\mathbb{Z}_2^{(0)} \times \mathbb{Z}_2^{(1)} \times \mathbb{Z}_{2, \text{swap}}^{(0)}$, where $\mathbb{Z}_{2, \text{swap}}^{(0)}$ is generated by the SPT entangler. The authors demonstrated that there is a mixed anomaly between $\mathbb{Z}_{2, \text{swap}}^{(0)}$ and $\mathbb{Z}_2^{(0)} \times \mathbb{Z}_2^{(1)}$ by computing the algebras of symmetry operators in various twisted sectors. This result also matches that in the continuum. In addition, the authors formulated a generalized Kennedy-Tasaki transformation on the lattice, which exchanges the above two cases. By using this transformation, the authors obtained concrete lattice models for two different SPT phases with a non-invertible duality symmetry. One of these models is the 2+1d cluster model, while the other model is new.

The analyses in this paper are very explicit and provide us with analytical tools to study non-invertible symmetries of 2+1d lattice models on firm ground. The results of the paper would have various potential applications, one of which is to study an interface between 2+1d SPT phases with non-invertible symmetries. There are also various interesting future directions as mentioned in the last section of the manuscript. Given the significance of the results and a clear potential for further developments, I would recommend this paper for publication.

Requested changes

I have a few minor questions and suggestions listed below.

  1. p.30, footnote 31, it would be helpful to comment on why "the generalized KT transformation is only unambiguous in the constrained Hilbert space $\widetilde{\mathcal{H}}$" because the operator $\mathsf{KT} = V\mathsf{D}V$ is well-defined also on the original tensor product Hilbert space $\mathcal{H}$.

  2. p.56, why does eq.(F.4) imply that "acting on an arbitrary state $|\psi\rangle$ with the entangler $V$ amounts to stacking a cluster state"? Do you use the fact that the diagonal entangler $V \otimes V$ acting on the doubled Hilbert space $\mathcal{H} \otimes \mathcal{H}$ is equivalent to the identity operator in an appropriate sense?

Please also find the following small typos.

  1. p.21, eq.(3.8), the dot between the first and second lines would not be necessary.

  2. p.40, the first paragraph of Section B.3, "can be view as" $\rightarrow$ "can be viewed as"

  3. p.42, eq.(C.7), "$\prod_{l \ni v} \sigma^x_v$" $\rightarrow$ "$\prod_{l \ni v} \sigma^x_l$"

  4. p.47, below eq.(D.9), "It exchanges the first and ..." would be a typo of "The first and ..."

  5. p.56, below eq.(F.4), "stacking $H$ with an $\mathbb{Z}_2^{(0)} \times \mathbb{Z}_2^{(1)}$ SPT" $\rightarrow$ "stacking $H$ with a $\mathbb{Z}_2^{(0)} \times \mathbb{Z}_2^{(1)}$ SPT"

  6. p.56, the first paragraph of Section F.2, "which exchanges exchange" $\rightarrow$ "which exchanges"

Recommendation

Publish (surpasses expectations and criteria for this Journal; among top 10%)

---

## Round 1 · Referee Report · Anonymous (Referee 2) · 2024-8-13

Report

The paper under review is an interesting and timely work, given the current impetus to understand the appearance and consequences of generalized and in particular non-invertible symmetries in lattice models.

This paper studies several instances of generalized symmetries with and without anomalies and in particular including non-invertible symmetries in models built from qubits. While most if not all of the phenomena discussed in this paper are known from a continuum field theory perspective it is still a very worthwhile exercise to illustrate it in the lattice setting.

In Sec. 2, the authors study a generalization of the well-known KW duality of the Ising model to 2+1d where they couple models with Z2 0 form symmetry and Z2 1 form symmetry. In general these models map into each other under gauging of Z2 0 form symmetry or conversely Z2 1 form symmetry. Hence there is a parameter subspace which is self dual under such gauging. This model hosts a non-invertible 2+1d KW self-duality symmetry which they study in detail.

In Sec 3, the paper demonstrates that the non-invertible symmetry thus obtained in Sec. 2 is related to a D8 invertible symmetry via gauging of the non-normal Z2 subgroup of D8. Specifically they discuss a 2+1d version of the Ashkin Teller model (in a restricted locus in parameter space) that realizes this symmetry and show that its gauging produces a model with a symmetry that corresponds to 2-representations of a 2-group.

In Sec. 4 the paper discusses yet another restricted region of their parent model that realizes a mixed anomaly between two Z2 0-form symmetries and a Z2 1-form symmetry and explore the consequences of this anomaly on the phase diagram .

In Sec. 5, the papers studies the 2+1d cluster SPT which is an SPT conventionally protected by Z2 0-form x Z2 1-form symmetry. The authors show that this splits into atleast two distinct SPTs protected by the non-invertible symmetry. This is a direct generalization of a recent work by a subset of the authors in one dimension lower.

I am happy to recommend this nice work for publication in Scipost however I have a couple of small questions/comments for the authors:

Requested changes

  1. It is claimed in the paper that 1.2 can be obtained by gauging the Z2 0-form symmetry in the Ising model. Isn’t this strictly speaking incorrect? Specifically when we gauge the Z2 0-form we do not land on a model that admits a tensor product Hilbert space. The dual Z2 1-form symmetry is topological in the sense used in the paper. A quick way to see that is that the product of ZZ around a plaquette is the identity before gauging but maps to a product of \sigma_z’s around a plaquette which must be enforced to be the identity as well.

  2. In this work, is it correct that both the SPTs realized reduce to the same SPT as a Z2 0-form x Z2 1-form SPT. Perhaps a more non-trivial example would be an SPT which trivializes on any invertible subgroup. Does this model realize such a phase?

  3. Can the mixed anomaly be viewed as a standard LSM anomaly involving a Z2 0-form symmetry and a Z2 1-form symmetry as the local representative of one of the Z2 symmetries contains a charged line of the 1-form symmetry?

Recommendation

Publish (easily meets expectations and criteria for this Journal; among top 50%)

---

## Editorial Decision

resubmitted